# Structured Adversarial Attack: Towards General Implementation and Better Interpretability

**Kaidi Xu**[1]* **Sijia Liu**[2]* **Pu Zhao**[1] **Pin-Yu Chen**[2] **Huan Zhang**[3] **Quanfu Fan**[2]
**Deniz Erdogmus**[1] **Yanzhi Wang**[1] **Xue Lin**[1]
[1]Northeastern University, USA
[2]MIT-IBM Watson AI Lab, IBM Research, USA
[3]University of California, Los Angeles, USA

## ABSTRACT

When generating adversarial examples to attack deep neural networks (DNNs), $\ell_p$ norm of the added perturbation is usually used to measure the similarity between original image and adversarial example. However, such adversarial attacks perturbing the raw input spaces may fail to capture structural information hidden in the input. This work develops a more general attack model, i.e., the structured attack (StrAttack), which explores group sparsity in adversarial perturbations by sliding a mask through images aiming for extracting key spatial structures. An ADMM (alternating direction method of multipliers)-based framework is proposed that can split the original problem into a sequence of analytically solvable subproblems and can be generalized to implement other attacking methods. Strong group sparsity is achieved in adversarial perturbations even with the same level of $\ell_p$-norm distortion ($p \in \{1, 2, \infty\}$) as the state-of-the-art attacks. We demonstrate the effectiveness of StrAttack by extensive experimental results on MNIST, CIFAR-10 and ImageNet. We also show that StrAttack provides better interpretability (i.e., better correspondence with discriminative image regions) through adversarial saliency map (Papernot et al., 2016b) and class activation map (Zhou et al., 2016). Our code is available at `https://github.com/KaidiXu/StrAttack`.

## 1 INTRODUCTION

Deep learning achieves exceptional successes in domains such as image recognition (He et al., 2016; Geifman & El-Yaniv, 2017), natural language processing (Hinton et al., 2012; Harwath et al., 2016), medical diagnostics (Chen et al., 2016; Shi et al., 2018) and advanced control (Silver et al., 2016; Fu et al., 2017). Recent studies (Szegedy et al., 2013; Goodfellow et al., 2014; Nguyen et al., 2015; Kurakin et al., 2016; Carlini & Wagner, 2017) show that DNNs are vulnerable to adversarial attacks implemented by generating adversarial examples, i.e., adding well-designed perturbations to original legal inputs. Delicately crafted adversarial examples can mislead a DNN to recognize them as any target image label, while the perturbations appears unnoticeable to human eyes. Adversarial attacks against

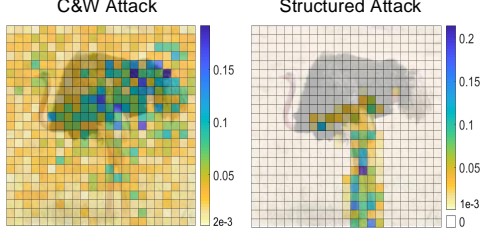

**Figure 1:** Group sparsity demonstrated in adversarial perturbations obtained by C&W attack and our StrAttack, where 'ostrich' is the original label, and 'unicycle' is the misclassified label. Here each group is a region of $13 \times 13 \times 3$ pixels and the strength of adversarial perturbations (through their $\ell_2$ norm) at each group is represented by heatmap. C&W attack perturbs almost all groups, while StrAttack yields strong group sparsity, with more semantic structure: the perturbed image region matches the feature of the target object, namely, the frame of the unicycle.

*Equal contribution

DNNs not only exist in theoretical models
but also pose potential security threats to the real world (Kurakin et al., 2016; Evtimov et al., 2017; Papernot et al., 2017). Several explanations are proposed to illustrate why there exist adversarial examples to DNNs based on hypotheses such as model linearity and data manifold (Goodfellow et al., 2014; Gilmer et al., 2018). However, little is known to their origins, and convincing explanations remain to be explored.

Besides achieving the goal of (targeted) mis-classification, an adversarial example should be as "similar" to the original legal input as possible to be stealthy. Currently, the similarity is measured by the $\ell_p$ norm ($p = 0, 1, 2, \infty$) of the added perturbation (Szegedy et al., 2013; Carlini & Wagner, 2017; Chen et al., 2017b;a), i.e., $\ell_p$ norm is being minimized when generating adversarial example. However, measuring the similarity between the original image and its adversarial example by $\ell_p$ norm is neither necessary nor sufficient (Sharif et al., 2018). Besides, no single measure can be perfect for human perceptual similarity (Carlini & Wagner, 2017) and such adversarial attacks may fail to capture key information hidden in the input such as spatial structure or distribution. Spurred by that, this work implements a new attack model i.e., *structured attack (StrAttack)* that imposes group sparsity on adversarial perturbations by extracting structures from the inputs. As shown in Fig. 1, we find that StrAttack identifies minimally sufficient regions that make attacks successful, but without incurring extra pixel-level perturbation power. The major contributions are summarized as below.

- (**Structure-driven attack**) This work is the first attempt towards exploring group-wise sparse structures when implementing adversarial attacks, but without losing $\ell_p$ distortion performance when compared to state-of-the-art attacking methods.

- (**Generality**) We show that the proposed attack model covers many norm-ball based attacks such as C&W (Carlini & Wagner, 2017) and EAD (Chen et al., 2017a).

- (**Efficient implementation**) We develop an efficient algorithm to generate structured adversarial perturbations by leveraging the alternating direction method of multipliers (ADMM). We show that ADMM splits the original complex problem into subproblems, each of which can be solved *analytically*. Besides, we show that ADMM can further be used to refine an arbitrary adversarial attack under the fixed sparse structure.

- (**Interpretability**) The generated adversarial perturbations demonstrate clear correlations and interpretations between original and target images. With the aid of adversarial saliency map (Papernot et al., 2016b) and class activation map (Zhou et al., 2016), we show that the obtained group-sparse adversarial patterns better shed light on the mechanisms of adversarial perturbations to fool DNNs.

**Related work** Many works studied norm-ball constrained adversarial attacks. For example, FGM (Goodfellow et al., 2014) and IFGSM (Kurakin et al., 2017) attack methods were proposed to maximize the classification error subject to $\ell_\infty$-norm based distortion constraints. Moreover, L-BFGS (Szegedy et al., 2013) and C&W (Carlini & Wagner, 2017) attacks found an adversarial example by minimizing its $\ell_2$-norm distortion. By contrast, JSMA (Papernot et al., 2016b) and one-pixel (Su et al., 2017) attacks attempted to generate adversarial examples by perturbing the minimum number of pixels, namely, minimizing the $\ell_0$ norm of adversarial perturbations. Different from the above norm-ball constrained attacks, some works (Karmon et al., 2018; Brown et al., 2017) crafted adversarial examples by adding noise patches. However, the resulting adversarial perturbations are no longer imperceptible to humans. Here we argue that imperceptibility could be important since it helps us to understand how/why DNNs are vulnerable to adversarial attacks while perturbing natural examples just by indistinguished adversarial noise.

In the aforementioned norm-ball constrained adversarial attacks, two extremely opposite principles have been applied: C&W attack (or $\ell_\infty$ attacks) seeks the minimum image-level distortion but allows to modify all pixels; one-pixel attack only perturbs a few pixels but suffers a high pixel-level distortion. Both attacking principles might lead to a high noise visibility due to perturbing too many pixels or perturbing a few pixels too much. In this work, we wonder if there exists a more effective attack that can be as successful as existing attacks but achieves a tradeoff between the perturbation power and the number of perturbed pixels. We will show that the proposed StrAttack is able to identify sparse perturbed regions that make attacks successful, but without incurring extra pixel-level

perturbations. It is also worth mentioning that one-pixel attack has much lower attack success rate on ImageNet than C&W attack and StrAttack.

In addition to adversarial attacks, many defense works have been proposed. Examples include defensive distillation (Papernot et al., 2016c) that distills the original DNN and introduces temperature into the softmax layer, random mask (Anonymous, 2019) that modifies the DNN structures by randomly removing certain neurons before training, adversarial training through enlarging the training dataset with adversarial examples, and robust adversarial training (Madry et al., 2017; Sinha et al., 2018) through the min-max optimization. It is commonly known that the robust adversarial training method ensures the strongest defense performance against adversarial attacks on MNIST and CIFAR-10. In this work, we will evaluate the effectiveness of StrAttack to three defense methods, a) defensive distillation (Papernot et al., 2016c), b) adversarial training via data augmentation (Tramèr et al., 2018) and c) robust adversarial training (Madry et al., 2017).

Although the adversarial attack and defense have attracted an increasing amount of attention, the visual explanation on adversarial perturbations is less explored since the distortion power is minimized and the resulting adversarial effects become imperceptible to humans. The work (Dong et al., 2017) attempted to understand how the internal representations of DNNs are affected by adversarial examples. However, only an ensemble-based attack was considered, which fails to distinguish the effectiveness of different norm-ball constrained adversarial attacks. Unlike (Dong et al., 2017), we employ the interpretability tools, adversarial saliency map (ASM) (Papernot et al., 2016b) and class activation map (CAM) (Zhou et al., 2016) to measure the effectiveness of different attacks in terms of their interpretability. Here ASM provides sensitivity analysis for pixel-level perturbation's impact on label classification, and CAM localizes class-specific image discriminative regions (Xiao et al., 2018). We will show that the sparse adversarial pattern obtained by StrAttack offers a great interpretability through ASM and CAM compared with other norm-ball constrained attacks.

## 2  STRUCTURED ATTACK: EXPLORE GROUP STRUCTURES FROM IMAGES

In the section, we introduce the concept of *StrAttack*, motivated by the question: 'what possible structures could adversarial perturbations have to fool DNNs?' Our idea is to divide an image into sub-groups of pixels and then penalize the corresponding group-wise sparsity. The resulting sparse groups encode minimally sufficient adversarial effects on local structures of natural images.

Let $\mathbf{\Delta} \in \mathbb{R}^{W \times H \times C}$ be an adversarial perturbation added to an original image $\mathbf{X}_0$, where $W \times H$ gives the spatial region, and $C$ is the depth, e.g., $C = 3$ for RGB images. To characterize the local structures of $\mathbf{\Delta}$, we introduce a *sliding mask* $\mathcal{M}$ with stride $S$ and size $r \times r \times C$. When $S = 1$, the mask moves one pixel at a time; When $S = 2$, the mask jumps 2 pixels at a time while sliding. By adjusting the stride $S$ and the mask size $r$, different group splitting schemes can be obtained. If $S < r$, the resulting groups will contain *overlapping* pixels. By contrast, groups will become *non-overlapped* when $S = r$.

A sliding mask $\mathcal{M}$ finally divides $\mathbf{\Delta}$ into a set of groups $\{\mathbf{\Delta}_{\mathcal{G}_{p,q}}\}$ for $p \in [P]$ and $q \in [Q]$, where $P = (W - r)/S + 1$, $Q = (H - r)/S + 1$, and $[n]$ denotes the integer set $\{1, 2, \ldots, n\}$. Given the groups $\{\mathbf{\Delta}_{\mathcal{G}_{p,q}}\}$, the group sparsity can be characterized through the following sparsity-inducing function (Yuan & Lin, 2006; Bach et al., 2012; Liu et al., 2015), motivated by the problem of group Lasso (Yuan & Lin, 2006):

$$g(\mathbf{\Delta}) = \sum_{p=1}^{P} \sum_{q=1}^{Q} \|\mathbf{\Delta}_{\mathcal{G}_{p,q}}\|_2, \tag{1}$$

where $\mathbf{\Delta}_{\mathcal{G}_{p,q}}$ denotes the set of pixels of $\mathbf{\Delta}$ indexed by $\mathcal{G}_{p,q}$, and $\| \cdot \|_2$ is the $\ell_2$ norm. We refer readers to Fig. A1 for an illustrative example of our concepts on groups and group sparsity.

## 3  STRUCTURED ADVERSARIAL ATTACK WITH ADMM

In this section, we start by proposing a general framework to generate prediction-evasive adversarial examples, where the adversary relies only on gradients of the loss function with respect to inputs of DNNs. Our model takes into account both commonly-used adversarial distortion metrics and the proposed group-sparsity regularization that encodes spatial structures in attacks. We show that the process of generating structured adversarial examples leads to an optimization problem that is

difficult to solve using the existing optimizers Adam (for C&W attack) and FISTA (for EAD attack) (Carlini & Wagner, 2017; Chen et al., 2017a). To circumvent this challenge, we develop an efficient optimization method via alternating direction method of multipliers (ADMM).

Given an original image $\mathbf{x}_0 \in \mathbb{R}^n$, we aim to design the optimal adversarial perturbation $\boldsymbol{\delta} \in \mathbb{R}^n$ so that the adversarial example $(\mathbf{x}_0 + \boldsymbol{\delta})$ misleads DNNs trained on natural images. Throughout this paper, we use vector representations of the adversarial perturbation $\boldsymbol{\Delta}$ and the original image $\mathbf{X}_0$ without loss of generality. A well designed perturbation $\boldsymbol{\delta}$ can be obtained by solving optimization problems of the following form,

$$\begin{aligned} \underset{\boldsymbol{\delta}}{\text{minimize}} \quad & f(\mathbf{x}_0 + \boldsymbol{\delta}, t) + \gamma D(\boldsymbol{\delta}) + \tau g(\boldsymbol{\delta}) \\ \text{subject to} \quad & (\mathbf{x}_0 + \boldsymbol{\delta}) \in [0, 1]^n, \ \|\boldsymbol{\delta}\|_\infty \leq \epsilon, \end{aligned} \quad (2)$$

where $f(\mathbf{x}, t)$ denotes the loss function for crafting adversarial example given a target class $t$, $D(\boldsymbol{\delta})$ is a distortion function that controls the perceptual similarity between a natural image and a perturbed image, $g(\boldsymbol{\delta}) = \sum_{p=1}^{P} \sum_{q=1}^{Q} \|\boldsymbol{\delta}_{\mathcal{G}_{p,q}}\|_2$ is given by (1), and $\|\cdot\|_p$ signifies the $\ell_p$ norm. In problem (2), the 'hard' constraints ensure the validness of created adversarial examples with $\epsilon$-tolerant perturbed pixel values. And the non-negative regularization parameters $\gamma$ and $\tau$ place our emphasis on the distortion of an adversarial example (to an original image) and group sparsity of adversarial perturbation. Tuning the regularization parameters will be discussed in Appendix F.

Problem (2) gives a quite general formulation for design of adversarial examples. If we remove the group-sparsity regularizer $g(\boldsymbol{\delta})$ and the $\ell_\infty$ constraint, problem (2) becomes the same as the C&W attack (Carlini & Wagner, 2017). More specifically, if we further set the distortion function $D(\boldsymbol{\delta})$ to the form of $\ell_0$, $\ell_2$ or $\ell_\infty$ norm, then we obtain C&W $\ell_0$, $\ell_2$ or $\ell_\infty$ attack. If $D(\boldsymbol{\delta})$ is specified by the elastic-net regularizer, then problem (2) becomes the formulation of EAD attack (Chen et al., 2017a).

In this paper, we specify the loss function of problem (2) as below, which yields the best known performance of adversaries (Carlini & Wagner, 2017),

$$f(\mathbf{x}_0 + \boldsymbol{\delta}, t) = c \cdot \max\{\max_{j \neq t} Z(\mathbf{x}_0 + \boldsymbol{\delta})_j - Z(\mathbf{x}_0 + \boldsymbol{\delta})_t, -\kappa\}, \quad (3)$$

where $Z(\mathbf{x})_j$ is the $j$th element of logits $Z(\mathbf{x})$, representing the output before the last softmax layer in DNNs, and $\kappa$ is a confidence parameter that is usually set to zero if the attack transferability is not much cared. We choose $D(\boldsymbol{\delta}) = \|\boldsymbol{\delta}\|_2^2$ for a fair comparison with the C&W $\ell_2$ adversarial attack. In this section, we assume that $\{\mathcal{G}_{p,q}\}$ are non-overlapping groups, i.e., $\mathcal{G}_{p,q} \cap \mathcal{G}_{p',q'} = \emptyset$ for $q \neq q'$ or $p \neq p'$. The overlapping case will be studied in the next section.

The presence of *multiple* non-smooth regularizers and 'hard' constraints make the existing optimizers Adam and FISTA (Carlini & Wagner, 2017; Chen et al., 2017a; Kingma & Ba, 2015; Beck & Teboulle, 2009) inefficient for solving problem (2). First, the subgradient of the objective function of problem (2) is difficult to obtain especially when $\{\mathcal{G}_{p,q}\}$ are overlapping groups. Second, it is impossible to compute the proximal operations required for FISTA with respect to all non-smooth regularizers and 'hard' constraints. Different from the existing work, we show that ADMM, a first-order operator splitting method, helps us to split the original complex problem (2) into a sequence of subproblems, each of which can be solved *analytically*.

We reformulate problem (2) in a way that lends itself to the application of ADMM,

$$\begin{aligned} \underset{\boldsymbol{\delta}, \mathbf{z}, \mathbf{w}, \mathbf{y}}{\text{minimize}} \quad & f(\mathbf{z} + \mathbf{x}_0) + \gamma D(\boldsymbol{\delta}) + \tau \sum_{i=1}^{PQ} \|\mathbf{y}_{\mathcal{D}_i}\|_2 + h(\mathbf{w}) \\ \text{subject to} \quad & \mathbf{z} = \boldsymbol{\delta}, \ \mathbf{z} = \mathbf{y}, \ \mathbf{z} = \mathbf{w}, \end{aligned} \quad (4)$$

where $\mathbf{z}$, $\mathbf{y}$ and $\mathbf{w}$ are newly introduced variables, for ease of notation let $\mathcal{D}_{(q-1)P+p} = \mathcal{G}_{p,q}$, and $h(\mathbf{w})$ is an indicator function with respect to the constraints of problem (2),

$$h(\mathbf{w}) = \begin{cases} 0 & \text{if } (\mathbf{x}_0 + \mathbf{w}) \in [0, 1]^n, \ \|\mathbf{w}\|_\infty \leq \epsilon, \\ \infty & \text{otherwise.} \end{cases} \quad (5)$$

ADMM is performed by minimizing the augmented Lagrangian of problem (4),

$$\begin{aligned} L(\mathbf{z}, \boldsymbol{\delta}, \mathbf{y}, \mathbf{w}, \mathbf{u}, \mathbf{v}, \mathbf{s}) = {} & f(\mathbf{z} + \mathbf{x}_0) + \gamma D(\boldsymbol{\delta}) + \tau \sum_{i=1}^{PQ} \|\mathbf{y}_{\mathcal{D}_i}\|_2 + h(\mathbf{w}) + \mathbf{u}^T(\boldsymbol{\delta} - \mathbf{z}) \\ & + \mathbf{v}^T(\mathbf{y} - \mathbf{z}) + \mathbf{s}^T(\mathbf{w} - \mathbf{z}) + \frac{\rho}{2}\|\boldsymbol{\delta} - \mathbf{z}\|_2^2 + \frac{\rho}{2}\|\mathbf{y} - \mathbf{z}\|_2^2 + \frac{\rho}{2}\|\mathbf{w} - \mathbf{z}\|_2^2, \end{aligned} \quad (6)$$

where $\mathbf{u}$, $\mathbf{v}$ and $\mathbf{s}$ are Lagrangian multipliers, and $\rho > 0$ is a given penalty parameter. ADMM splits all of optimization variables into *two* blocks and adopts the following iterative scheme,

$$\{\boldsymbol{\delta}^{k+1}, \mathbf{w}^{k+1}, \mathbf{y}^{k+1}\} = \underset{\boldsymbol{\delta}, \mathbf{w}, \mathbf{y}}{\arg\min} \, L(\boldsymbol{\delta}, \mathbf{z}^k, \mathbf{w}, \mathbf{y}, \mathbf{u}^k, \mathbf{v}^k, \mathbf{s}^k), \tag{7}$$

$$\mathbf{z}^{k+1} = \underset{\mathbf{z}}{\arg\min} \, L(\boldsymbol{\delta}^{k+1}, \mathbf{z}, \mathbf{w}^{k+1}, \mathbf{y}^{k+1}, \mathbf{u}^k, \mathbf{v}^k, \mathbf{s}^k), \tag{8}$$

$$\begin{cases} \mathbf{u}^{k+1} = \mathbf{u}^k + \rho(\boldsymbol{\delta}^{k+1} - \mathbf{z}^{k+1}), \\ \mathbf{v}^{k+1} = \mathbf{v}^k + \rho(\mathbf{y}^{k+1} - \mathbf{z}^{k+1}), \\ \mathbf{s}^{k+1} = \mathbf{s}^k + \rho(\mathbf{w}^{k+1} - \mathbf{z}^{k+1}), \end{cases} \tag{9}$$

where $k$ is the iteration index, steps (7)-(8) are used for updating primal variables, and the last step (9) is known as the dual update step. We emphasize that the crucial property of the proposed ADMM approach is that, as we demonstrate in Proposition 1, the solution to problem (7) can be found in parallel and exactly.

**Proposition 1** *When $D(\boldsymbol{\delta}) = \|\boldsymbol{\delta}\|_2^2$, the solution to problem (7) is given by*

$$\boldsymbol{\delta}^{k+1} = \frac{\rho}{\rho + 2\gamma} \mathbf{a}, \tag{10}$$

$$[\mathbf{w}^{k+1}]_i = \begin{cases} \min\{1 - [\mathbf{x}_0]_i, \epsilon\} & b_i > \min\{1 - [\mathbf{x}_0]_i, \epsilon\} \\ \max\{-[\mathbf{x}_0]_i, -\epsilon\} & b_i < \max\{-[\mathbf{x}_0]_i, -\epsilon\} \quad \text{for } i \in [n], \\ b_i & \text{otherwise}, \end{cases} \tag{11}$$

$$[\mathbf{y}^{k+1}]_{\mathcal{D}_i} = \left(1 - \frac{\tau}{\rho \| [\mathbf{c}]_{\mathcal{D}_i} \|_2}\right)_+ [\mathbf{c}]_{\mathcal{D}_i}, \; i \in [PQ], \tag{12}$$

*where $\mathbf{a} := \mathbf{z}^k - \mathbf{u}^k/\rho$, $\mathbf{b} := \mathbf{z}^k - \mathbf{s}^k/\rho$, $\mathbf{c} := \mathbf{z}^k - \mathbf{v}^k/\rho$, $(x)_+ = x$ if $x \geq 0$ and $0$ otherwise, $[\mathbf{x}]_i$ denotes the $i$th element of $\mathbf{x}$, and $[\mathbf{x}]_{\mathcal{D}_i}$ denotes the sub-vector of $\mathbf{x}$ indexed by $\mathcal{D}_i$.*

**Proof:** See Appendix B. $\qquad\qquad\qquad\qquad\qquad\qquad\qquad\qquad\qquad\qquad\qquad\qquad\qquad\qquad$ $\square$

It is clear from Proposition 1 that introducing auxiliary variables does not increase the computational complexity of ADMM since (10)-(12) can be solved in parallel. Moreover, if another distortion metric (different from $D(\boldsymbol{\delta}) = \|\boldsymbol{\delta}\|_2^2$) is used, then ADMM only changes at the $\boldsymbol{\delta}$-step (10).

We next focus on the $\mathbf{z}$-minimization step (8), which can be equivalently transformed into

$$\underset{\mathbf{z}}{\text{minimize}} \quad f(\mathbf{x}_0 + \mathbf{z}) + \frac{\rho}{2}\|\mathbf{z} - \mathbf{a}'\|_2^2 + \frac{\rho}{2}\|\mathbf{z} - \mathbf{b}'\|_2^2 + \frac{\rho}{2}\|\mathbf{z} - \mathbf{c}'\|_2^2, \tag{13}$$

where $\mathbf{a}' := \boldsymbol{\delta}^{k+1} + \mathbf{u}^k/\rho$, $\mathbf{b}' := \mathbf{w}^{k+1} + \mathbf{s}^k/\rho$, and $\mathbf{c}' := \mathbf{y}^{k+1} + \mathbf{v}^k/\rho$. We recall that attacks studied in this paper belongs to 'first-order' adversaries (Madry et al., 2017), which only have access to gradients of the loss function $f$. Spurred by that, we solve problem (13) via a linearization technique that is commonly used in stochastic/online ADMM (Ouyang et al., 2013; Suzuki, 2013; Liu et al., 2018) or linearized ADMM (Boyd et al., 2011; Liu et al., 2017). Specifically, we replace the function $f$ with its first-order Taylor expansion at the point $\mathbf{z}^k$ by adding a Bregman divergence term $(\eta_k/2)\|\mathbf{z} - \mathbf{z}^k\|_2^2$. As a result, problem (13) becomes

$$\underset{\mathbf{z}}{\text{minimize}} \quad (\nabla f(\mathbf{z}^k + \mathbf{x}_0))^T(\mathbf{z} - \mathbf{z}^k) + \frac{\eta_k}{2}\|\mathbf{z} - \mathbf{z}^k\|_2^2 + \frac{\rho}{2}\|\mathbf{z} - \mathbf{a}'\|_2^2 \\ + \frac{\rho}{2}\|\mathbf{z} - \mathbf{b}'\|_2^2 + \frac{\rho}{2}\|\mathbf{z} - \mathbf{c}'\|_2^2, \tag{14}$$

where $1/\eta_k > 0$ is a given decaying parameter, e.g., $\eta_k = \alpha\sqrt{k}$ for some $\alpha > 0$, and the Bregman divergence term stabilizes the convergence of $\mathbf{z}$-minimization step. It is clear that problem (14) yields a quadratic program with the closed-form solution

$$\mathbf{z}^{k+1} = (1/(\eta_k + 3\rho)) \left(\eta_k \mathbf{z}^k + \rho\mathbf{a} + \rho\mathbf{b} + \rho\mathbf{c} - \nabla f(\mathbf{z}^k + \mathbf{x}_0)\right). \tag{15}$$

In summary, the proposed ADMM algorithm alternatively updates (7)-(9), which yield closed-form solutions given by (10)-(12) and (15). The convergence of linearized ADMM for nonconvex optimization was recently proved by (Liu et al., 2017), and thus provides theoretical validity of our approach. Compared to the existing solver for generation of adversarial examples (Carlini & Wagner, 2017; Papernot et al., 2016b), our algorithm offers two main benefits, *efficiency* and *generality*. That is, the computations for every update step are efficiently carried out, and our approach can be applicable to a wide class of attack formulations.

## 4 OVERLAPPING GROUP AND REFINED STRATTACK

In this section, we generalize our proposed ADMM solution framework to the case of generating adversarial perturbations with *overlapping* group structures. We then turn to an attack refining model under *fixed* sparse structures. We will show that both extensions can be unified under the ADMM framework. In particular, the refined approach will allow us to gain deeper insights on the structural effects on adversarial perturbations.

### 4.1 OVERLAPPING GROUP STRUCTURE

We recall that groups $\{\mathcal{D}_i\}$ (also denoted by $\{\mathcal{G}_{p,q}\}$) studied in Sec. 3 could be overlapped with each other; see an example in Fig. A1. Therefore, $\{\mathcal{D}_i\}$ is in general a *cover* rather than a partition of $[n]$. To address the challenge in coupled group variables, we introduce multiple copies of the variable $\mathbf{y}$ in problem (4), and achieve the following modification

$$
\begin{array}{ll}
\underset{\boldsymbol{\delta}, \mathbf{z}, \mathbf{w}, \{\mathbf{y}_i\}}{\text{minimize}} & f(\mathbf{z} + \mathbf{x}_0) + \gamma D(\boldsymbol{\delta}) + \tau \sum_{i=1}^{PQ} \|\mathbf{y}_{i,\mathcal{D}_i}\|_2 + h(\mathbf{w}) \\
\text{subject to} & \mathbf{z} = \boldsymbol{\delta}, \ \mathbf{z} = \mathbf{w}, \ \mathbf{z} = \mathbf{y}_i, \quad i \in [PQ],
\end{array} \tag{16}
$$

where compared to problem (4), there exist $PQ$ variables $\mathbf{y}_i \in \mathbb{R}^n$ for $i \in [PQ]$, and $\mathbf{y}_{i,\mathcal{D}_i}$ denotes the subvector of $\mathbf{y}_i$ with indices given by $\mathcal{D}_i$. It is clear from (16) that groups $\{\mathcal{D}_i\}$ become *non-overlapped* since each of them lies in a different copy $\mathbf{y}_i$. The ADMM algorithm for solving problem (16) maintains a similar procedure as (7)-(9) except $\mathbf{y}$-step (12) and $\mathbf{z}$-step (15); see Proposition 2.

**Proposition 2** *Given the same condition of Proposition 1, the ADMM solution to problem (16) involves the $\boldsymbol{\delta}$-step same as (10), the $\mathbf{w}$-step same as (11), and two modified $\mathbf{y}$- and $\mathbf{z}$-steps,*

$$
\left\{
\begin{array}{l}
\left[\mathbf{y}_i^{k+1}\right]_{\mathcal{D}_i} = \left(1 - \frac{\tau}{\rho \|[\mathbf{c}_i]_{\mathcal{D}_i}\|_2}\right)_+ [\mathbf{c}_i]_{\mathcal{D}_i} \\
\left[\mathbf{y}_i^{k+1}\right]_{[n]/\mathcal{D}_i} = [\mathbf{c}_i]_{[n]/\mathcal{D}_i}
\end{array}
\right. , \ \textit{for } i \in [PQ], \tag{17}
$$

$$
\mathbf{z}^{k+1} = \left(1/\left(\eta_k + 2\rho + PQ\rho\right)\right) \left(\eta_k \mathbf{z}^k + \rho \mathbf{a}' + \rho \mathbf{b}' + \rho \sum_{i=1}^{PQ} \mathbf{c}_i' - \nabla f(\mathbf{z}^k + \mathbf{x}_0)\right), \tag{18}
$$

*where $\mathbf{c}_i := \mathbf{z}^k - \mathbf{v}_i^k/\rho$, $\mathbf{v}_i$ is the Lagrangian multiplier associated with equality constraint $\mathbf{y}_i = \mathbf{z}$, similar to (9) we obtain $\mathbf{v}_i^{k+1} = \mathbf{v}_i^k + \rho(\mathbf{y}^{k+1} - \mathbf{z}^{k+1})$, $[n]/\mathcal{D}_i$ denotes the difference of sets $[n]$ and $\mathcal{D}_i$, $\mathbf{a}'$ and $\mathbf{b}'$ have been defined in (13), and $\mathbf{c}_i' = \mathbf{y}_i^{k+1} + \mathbf{v}_i^k/\rho$.*

**Proof**: See Appendix C. □

We note that updating $PQ$ variables $\{\mathbf{y}_i\}$ is decomposed as shown in (17). However, the side effect is the need of $PQ$ times more storage space than the $\mathbf{y}$-step (12) when groups are non-overlapped.

### 4.2 REFINED STRATTACK UNDER FIXED SPARSE PATTERN

The approaches proposed in Sec. 3 and Sec. 4.1 help us to identify structured sparse patterns in adversarial perturbations. This section presents a method to refine structured attacks under fixed group sparse patterns. Let $\boldsymbol{\delta}^*$ denote the solution to problem (2) solved by the proposed ADMM method. We define a $\sigma$-sparse perturbation $\boldsymbol{\delta}$ via $\boldsymbol{\delta}^*$,

$$
\delta_i = 0 \ \text{if } \delta_i^* \leq \sigma, \ \text{for any } i \in [n], \tag{19}
$$

where a hard thresholding operator is applied to $\boldsymbol{\delta}^*$ with tolerance $\sigma$. Our refined model imposes the fixed $\sigma$-sparse structure (19) into problem (2). This leads to

$$
\begin{array}{ll}
\underset{\boldsymbol{\delta}}{\text{minimize}} & f(\mathbf{x}_0 + \boldsymbol{\delta}) + \gamma D(\boldsymbol{\delta}) \\
\text{subject to} & (\mathbf{x}_0 + \boldsymbol{\delta}) \in [0, 1]^n, \ \|\boldsymbol{\delta}\|_\infty \leq \epsilon \\
& \delta_i = 0, \ \text{if } i \in \mathcal{S}_\sigma,
\end{array} \tag{20}
$$

where $\mathcal{S}_\sigma$ is defined by (19), i.e., $\mathcal{S}_\sigma := \{j \mid \delta_j^* \leq \sigma, j \in [n]\}$. Compared to problem (2), the group-sparse penalty function is eliminated as it has been known as *a priori*. With the priori knowledge of group sparsity, problem (20) is formulated to optimize and refine the non-zero groups, thus achieving better performance on highlighting and exploring the perturbation structure. Problem (20) can be solved using ADMM, and its solution is presented in Proposition 3.

**Proposition 3** *The ADMM solution to problem (20) is given by*

$$[\boldsymbol{\delta}^{k+1}]_i = \begin{cases} 0 & i \in \mathcal{S}_\sigma \\ \min\{1 - [\mathbf{x}_0]_i, \epsilon\} & \frac{\rho}{2\gamma+\rho}a_i > \min\{1 - [\mathbf{x}_0]_i, \epsilon\}, i \notin \mathcal{S}_\sigma \\ \max\{-[\mathbf{x}_0]_i, -\epsilon\} & \frac{\rho}{2\gamma+\rho}a_i < \max\{-[\mathbf{x}_0]_i, -\epsilon\}, i \notin \mathcal{S}_\sigma \\ \frac{\rho}{2\gamma+\rho}a_i & otherwise, \end{cases} \tag{21}$$

$$[\mathbf{z}^{k+1}]_i = \begin{cases} 0 & i \in \mathcal{S}_\sigma \\ 1/(\eta_k + \rho)\left[\eta_k[\mathbf{z}^k]_i + \rho[\mathbf{a}']_i - [\nabla f(\mathbf{z}^k + \mathbf{x}_0)]_i\right] & i \notin \mathcal{S}_\sigma, \end{cases} \tag{22}$$

*for $i \in [n]$, where $\mathbf{z} = \boldsymbol{\delta}$ is the introduced auxiliary variable similar to (4), $\mathbf{a} := \boldsymbol{\delta}^{k+1} - \mathbf{u}^k/\rho$, $\mathbf{a}' := \boldsymbol{\delta}^{k+1} + \mathbf{u}^k/\rho$, $\mathbf{u}^{k+1} = \mathbf{u}^k + \rho(\boldsymbol{\delta}^{k+1} - \mathbf{z}^{k+1})$, and $\rho$ and $\eta_k$ have been defined in (6) and (14). The ADMM iterations can be initialized by $\boldsymbol{\delta}^*$, the known solution to problem (2).*

**Proof**: See Appendix D. □

## 5 EMPIRICAL PERFORMANCE OF STRATTACK

We evaluate the performance of the proposed StrAttack on three image classification datasets, MNIST (Lecun et al., 1998), CIFAR-10 (Krizhevsky & Hinton, 2009) and ImageNet (Deng et al., 2009). To make fair comparison with the C&W $\ell_2$ attack (Carlini & Wagner, 2017), we use $\ell_2$ norm as the distortion function $D(\boldsymbol{\delta}) = \|\boldsymbol{\delta}\|_2^2$. And we also compare with FGM (Goodfellow et al., 2014) and IFGSM $\ell_2$ attacks (Kurakin et al., 2017) as a reference. We evaluate attack success rate (ASR)[1] as well as $\ell_p$ distortion metrics for $p \in \{0, 1, 2, \infty\}$. The detailed experiment setup is presented in Appendix F. Our code is available at `https://github.com/KaidiXu/StrAttack`.

For each attack method on MNIST or CIFAR-10, we choose 1000 original images from the test dataset as source and each image has 9 target labels. So a total of 9000 adversarial examples are generated for each attack method. On ImageNet, each attack method tries to craft 900 adverdarial examples with 100 random images from the test dataset and 9 random target labels for each image.

Fig. 2 compares adversarial examples generated by StrAttack and C&W attack on each dataset. We observe that the perturbation of the C&W attack has poor group sparsity, i.e., many non-zeros groups with small magnitudes. However, the ASR of the C&W attack is quite sensitive to these small perturbations. As applying a threshold to have the same $\ell_0$ norm as our attack, we find that only 6.7% of adversarial examples generated from C&W attack remain valid. By contrast, StrAttack is able to highlight the most important group structures (local regions) of adversarial perturbations without attacking other pixels. For example, StrAttack misclassifies a natural image (4 in MNIST) as an incorrect label 3. That is because the pixels that appears in the structure of 3 are more significantly perturbed by our attack; see the top right plots of Fig. 2. Furthermore, the 'goose-sorrel' example shows that misclassification occurs when we just perturb a small number of non-sparse group regions on goose's head, which is more consistent with human perception. We refer readers to Appendix G for more results.

By quatitatively analysis, we report $\ell_p$ norms and ASR in Table 1 for $p \in \{0, 1, 2, \infty\}$. We show that StrAttack perturbs much fewer pixels (smaller $\ell_0$ norm), but it is comparable to or even better than other attacks in terms of $\ell_1$, $\ell_2$, and $\ell_\infty$ norms. Specifically, the FGM attack yields the worst performance in both ASR and $\ell_p$ distortion. On MNIST and CIFAR-10, StrAttack outperforms other attacks in $\ell_0$, $\ell_1$ and $\ell_\infty$ distortion. On ImageNet, StrAttack outperforms C&W attack in $\ell_0$ and $\ell_1$ distortion. Since the C&W attacking loss directly penalizes the $\ell_2$ norm, it often causes smaller $\ell_2$ distortion than StrAttack. We also observe that the overlapping case leads to the adversarial perturbation of less sparsity (in terms of $\ell_0$ norm) compared to the non-overlapping case. This is not surprising, since the sparsity of the overlapping region is controlled by at least two groups. However, compared to C&W attack, the use of overlapping groups in StrAttack still yields sparser perturbations. Unless specified otherwise, we focus on the case of non-overlapping groups to generate the most sparse adversarial perturbations. We highlight that although a so-called one-pixel attack (Su et al., 2017) also yields very small $\ell_0$ norm, it is at the cost of very large $\ell_\infty$ distortion. Unlike one-pixel attack, StrAttack achieves the sparsity *without losing* the performance of $\ell_\infty$, $\ell_1$ and $\ell_2$ distortion.

Furthermore, we compare the performance of StrAttack with the C&W $\ell_\infty$ attack and IFGSM while attacking the robust model (Madry et al., 2017) on MNIST. We remark that all the considered attack

---

[1]The percentage of adversarial examples that successfully fool DNNs.

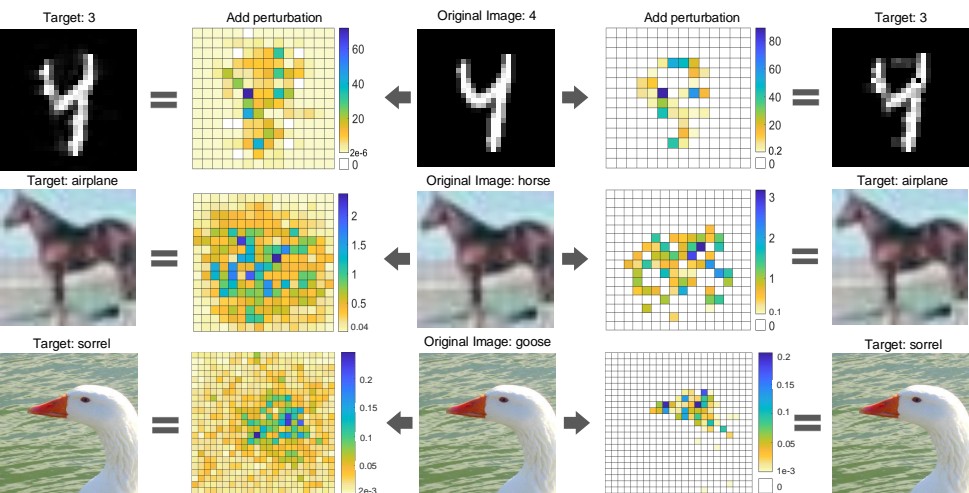

**Figure 2:** C&W attack vs StrAttack. Here each grid cell represents a $2 \times 2$, $2 \times 2$, and $13 \times 13$ small region in MNIST, CIFAR-10 and ImageNet, respectively. The group sparsity of perturbation is represented by heatmap. The colors on heatmap represent average absolute value of distortion scale to $[0, 255]$. The left two columns correspond to results of using C&W attack. The right two columns show results of StrAttack.

**Table 1:** Adversarial attack success rate (ASR) and $\ell_p$ distortion values for various attacks.

| Data Set | Attack Method | Best Case[*] | | | | | Average Case[*] | | | | | Worst Case[*] | | | | |
|---|---|---|---|---|---|---|---|---|---|---|---|---|---|---|---|---|
| | | ASR | $\ell_0$ | $\ell_1$ | $\ell_2$ | $\ell_\infty$ | ASR | $\ell_0$ | $\ell_1$ | $\ell_2$ | $\ell_\infty$ | ASR | $\ell_0$ | $\ell_1$ | $\ell_2$ | $\ell_\infty$ |
| MNIST | FGM | 99.3 | 456.5 | 28.2 | 2.32 | 0.57 | 35.8 | 466 | 39.4 | 3.17 | 0.717 | 0 | N.A.[**] | N.A. | N.A. | N.A. |
| | IFGSM | 100 | 549.5 | 18.3 | 1.57 | 0.4 | 100 | 588 | 30.9 | 2.41 | 0.566 | 99.8 | 640.4 | 50.98 | 3.742 | 0.784 |
| | C&W | 100 | 479.8 | 13.3 | 1.35 | 0.397 | 100 | 493.4 | 21.3 | **1.9** | 0.528 | 99.7 | 524.3 | 29.9 | **2.45** | 0.664 |
| | StrAttack | 100 | **73.2** | 10.9 | 1.51 | **0.384** | 100 | **119.4** | 18.05 | 2.16 | **0.47** | 100 | **182.0** | 26.9 | 2.81 | **0.5** |
| | +overlap | 100 | 84.4 | **9.2** | **1.32** | 0.401 | 100 | 157.4 | **16.2** | 1.95 | 0.508 | 100 | 260.9 | **22.9** | 2.501 | 0.653 |
| CIFAR-10 | FGM | 98.5 | 3049 | 12.9 | 0.389 | 0.046 | 44.1 | 3048 | 34.2 | 0.989 | 0.113 | 0.2 | 3071 | 61.3 | 1.76 | 0.194 |
| | IFGSM | 100 | 3051 | 6.22 | 0.182 | 0.02 | 100 | 3051 | 13.7 | 0.391 | 0.0433 | 100 | 3060 | 22.9 | 0.655 | 0.075 |
| | C&W | 100 | 2954 | 6.03 | 0.178 | **0.019** | 100 | 2956 | 12.1 | 0.347 | **0.0364** | 99.9 | 3070 | 16.8 | **0.481** | 0.0536 |
| | StrAttack | 100 | **264** | **3.33** | 0.204 | 0.031 | 100 | **487** | 7.13 | 0.353 | 0.050 | 100 | **772** | **12.5** | 0.563 | 0.075 |
| | +overlap | 100 | 295 | 3.35 | **0.169** | 0.029 | 100 | 562 | **7.05** | **0.328** | 0.047 | 100 | 920 | 12.9 | 0.502 | 0.063 |
| ImageNet | FGM | 12 | 264917 | 152 | 0.477 | **0.0157** | 2 | 263585 | 51.3 | **0.18** | **0.00614** | 0 | N.A. | N.A. | N.A. | N.A. |
| | IFGSM | 100 | 267079 | 299.32 | 0.9086 | 0.02964 | 100 | 267293 | 723 | 2.2 | 0.0792 | 98 | 267581 | 1378 | 4.22 | 0.158 |
| | C&W | 100 | 267916 | 127 | **0.471** | 0.016 | 100 | 263140 | 198 | 0.679 | 0.03 | 100 | 265212 | 268 | **0.852** | **0.041** |
| | StrAttack | 100 | **14462** | **55.2** | 0.719 | 0.058 | 100 | **52328** | **152** | 1.06 | 0.075 | 100 | **80722** | **197** | 1.35 | 0.122 |

[*] Please refer to Appendix F for the definition of best case, best case and worst case.
[**] N.A. means not available in the case of zero ASR, +overlap means structured attack with overlapping groups.

methods are performed under the same $\ell_\infty$-norm based distortion constraint with an upper bound $\epsilon \in \{0.1, 0.2, 0.3, 0.4\}$. Here we obtain a (refined) StrAttack subject to $\|\boldsymbol{\delta}\|_\infty \leq \epsilon$ by solving problem (20) at $\gamma = 0$. In Table 2, we demonstrate the ASR and the number of perturbed pixels for various attacks over 5000 (untargeted) adversarial examples. The ASR define as the proportion of the final perturbation results less than given $\epsilon \in \{0.1, 0.2, 0.3, 0.4\}$ bound over number of test images. Here an successful attack is defined by an attack that can fool DNNs and meets the $\ell_\infty$ distortion constraint. As we can see, StrAttack can achieve the similar ASR compared to other attack methods, however, it perturbs a much less number of pixels. Next, we evaluate the performance of StrAttack against two defense mechanisms: defensive distillation (Papernot et al., 2016c) and adversarial training (Tramèr et al., 2018). We observe that StrAttack is able to break the two defense methods with 100% ASR. More details are provided in Appendix H.

Lastly, we evaluate the transferability of StrAttack from Inception V3 (Szegedy et al., 2016) to other network models including Inception V2, Inception V4 (Szegedy et al., 2017), ResNet 50, ResNet 152 (He et al., 2016), DenseNet 121 and DenseNet 161 (Huang et al., 2017). For comparison, we also present the transferbility of IFGSM and C&W. This experiment is performed under 1000 (target) adversarial examples on ImageNet[2]. It can be seen from in Table 3 that StrAttack yields the largest attack success rate while transferring to almost every network model.

---

[2]We follow the experiment setting in (Su et al., 2018), where the transferability is evaluated by the target class top-5 success rate at each transferred model.

**Table 2:** Attack success rate (ASR) and $\ell_0$ norm of adversarial perturbations for various attacks against robust adversarial training based defense on MNIST.

|  | ASR at $\epsilon = 0.1$ | ASR at $\epsilon = 0.2$ | ASR at $\epsilon = 0.3$ | ASR at $\epsilon = 0.4$ | $\ell_0$ |
|---|---|---|---|---|---|
| IFGSM | 0.01 | 0.02 | 0.09 | 0.94 | 654 |
| C&W $\ell_\infty$ attack | 0.01 | 0.02 | 0.10 | 0.96 | 723 |
| StrAttack | 0.01 | 0.02 | 0.10 | 0.99 | 279 |

**Table 3:** Comparison of transferability of different attacks over 6 ImageNet models.

|  | Incept V2 | Incept V4 | ResNet50 | ResNet152 | DenseNet121 | DenseNet161 |
|---|---|---|---|---|---|---|
| IFGSM | 0.27 | 0.22 | **0.27** | 0.19 | 0.16 | 0.19 |
| C&W | 0.25 | 0.24 | 0.23 | 0.23 | 0.15 | 0.15 |
| StrAttack | **0.28** | **0.27** | 0.25 | **0.25** | **0.26** | **0.25** |

## 6 STRATTACK OFFERS BETTER INTERPRETABILITY

In this section, we evaluate the effects of structured adversarial perturbations on image classification through adversarial saliency map (ASM) (Papernot et al., 2016b) and class activation map (CAM) (Zhou et al., 2016). Here we recall that ASM measures the impact of pixel-level perturbations on label classification, and CAM localizes class-specific image discriminative regions that we use to visually explain adversarial perturbations (Xiao et al., 2018). We will show that compared to C&W attack, StrAttack meets better interpretability in terms of (a) a higher ASM score and (b) a tighter connection with CAM, where the metric (a) implies interpretability at a micro-level, namely, perturbing pixels with largest impact on image classification, and the metric (b) demonstrates interpretability at a macro-level, namely, perturbations can be mapped to the most discriminative image regions localized by CAM.

Given an input image $\mathbf{x}_0$ and a target class $t$, let $\mathrm{ASM}(\mathbf{x}_0, t) \in \mathbb{R}^d$ denote ASM scores for every pixel of $\mathbf{x}_0$ corresponding to $t$. We elaborate on the mathematical definition of ASM in Appendix E. Generally speaking, the $i$th element of $\mathrm{ASM}(\mathbf{x}_0, t)$, denoted by $\mathrm{ASM}(\mathbf{x}_0, t)[i]$, measures how much the classification score with respect to the target label $t$ will increase and that with respect to the original label $t_0$ will decrease if a perturbation is added to the pixel $i$. With the aid of ASM, we then define a Boolean map $\mathbf{B}_{\mathrm{ASM}} \in \mathbb{R}^d$ to encode the regions of $\mathbf{x}_0$ most sensitive to targeted adversarial attacks, where $\mathbf{B}_{\mathrm{ASM}}(i) = 1$ if $\mathrm{ASM}(\mathbf{x}_0, t) > \nu$, and 0 otherwise. Here $\nu$ is a given threshold to highlight the most sensitive pixels. we then define the interpretability score (IS) via ASM,

$$\mathrm{IS}(\boldsymbol{\delta}) = \|\mathbf{B}_{\mathrm{ASM}} \circ \boldsymbol{\delta}\|_2 / \|\boldsymbol{\delta}\|_2, \tag{23}$$

where $\circ$ is the element-wise product. The rationale behind (23) is that $\mathrm{IS}(\boldsymbol{\delta}) \to 1$ if the sensitive region identified by ASM perfectly predicts the locations of adversarial perturbations. By contrast, if $\mathrm{IS}(\boldsymbol{\delta}) \to 0$, then adversarial perturbations cannot be interpreted by ASM. In Fig. 3(a), we compare IS of our proposed attack with C&W attack versus the threshold $\nu$, valued by different percentiles of ASM scores. We obsrve that our attack outperforms C&W attack in terms of IS, since the former is able to extract important local structures of images by penalizing the group sparsity of adversarial perturbations. It seems that our improvement is not significant. However, StrAttack just perturbs very few pixels to obtain this benefit, leading to perturbations with more semantic structure; see Fig. 3(b) for an illustrative example.

Besides ASM, we show that the effect of adversarial perturbations can be visually explained through the class-specific discriminative image regions localized by CAM (Zhou et al., 2016). In Fig. 3(c), we illustrate CAM and demonstrate the differences between our attack and C&W in terms of their connections to the most discriminative regions of $\mathbf{x}_0$ with label $t_0$. We observe that the mechanism of StrAttack can be better interpreted from CAM: only a few adversarial perturbations are needed to suppress the feature of the original image with the true label. By replacing ASM with CAM, we can similarly compute IS in (23) averaged over $500$ examples on ImageNet, yielding $0.65$ for C&W attack and $0.77$ for our attack. More examples of ASM and CAM can be viewed in Appendix E.

To better interpret the mechanism of adversarial examples, we study adversarial attacks on some complex images, where the objects of the original and target labels exist simultaneously as shown in Fig. 4. It can be visualized from CAM that both C&W attack and StrAttack yields similar adversarial effects on natural images: Adversarial perturbations are used to suppress the most discriminative

region with respect to the true label, and simultaneously promotes the discriminative region of the target label. The former principle is implied by the location of perturbed regions and $C(\mathbf{x}_0, t_0)$ in Fig. 4, and the latter can be seen from $C(\mathbf{x}_{\text{CW}}, t)$ or $C(\mathbf{x}_{\text{Str}}, t)$ against $C(\mathbf{x}_0, t)$. However, compared to C&W attack, StrAttack perturbs much less but 'right' pixels which have better correspondence with class-specific discriminative image regions localized by CAM.

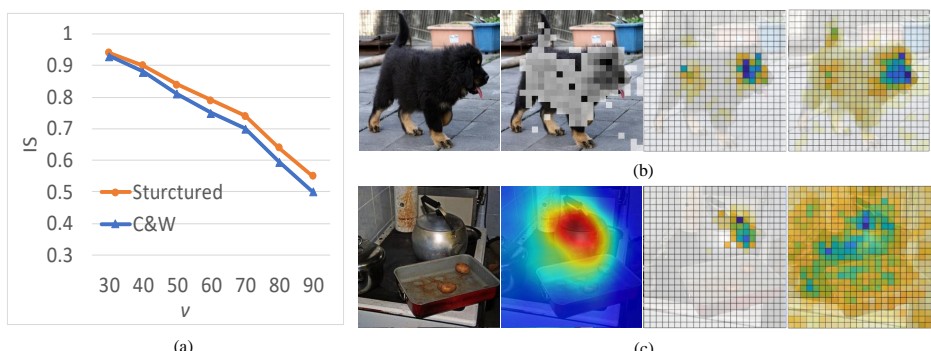

**Figure 3:** Interpretabilicy comparison of StrAttack and C&W attack. (a) ASM-based IS vs $\nu$, given from the 30th percentile to the 90th percentile of ASM scores. (b) Overlay ASM and $\mathbf{B}_{\text{ASM}} \circ \boldsymbol{\delta}$ on top of image with the true label 'Tibetan Mastiff' and the target label 'streetcar'. From left to right: original image, ASM (darker color represents larger value of ASM score), $\mathbf{B}_{\text{ASM}} \circ \boldsymbol{\delta}$ under StrAttack, and $\mathbf{B}_{\text{ASM}} \circ \boldsymbol{\delta}$ under C&W attack. Here $\nu$ in $\mathbf{B}_{\text{ASM}}$ is set by the 90th percentile of ASM scores. (c) From left to right: original image with true label 'stove', CAM of 'stove', and perturbations with target label 'water ouzel' under StrAttack and C&W.

## 7 CONCLUSION

This work explores group-wise sparse structures when implementing adversarial attacks. Different from previous works that use $\ell_p$ norm to measure the similarity between an original image and an adversarial example, this work incorporates group-sparsity regularization into the problem formulation of generating adversarial examples and achieves strong group sparsity in the obtained adversarial perturbations. Leveraging ADMM, we develop an efficient implementation to generate structured adversarial perturbations, which can be further used to refine an arbitrary adversarial attack under fixed group sparse structures. The proposed ADMM framewrok is general enough for implementing many state-of-the-art attacks. We perform extensive experiments using MNIST, CIFAR-10 and ImageNet datasets, showing that our structured adversarial attack (StrAttack) is much stronger than the existing attacks and its better interpretability from group sparse structures aids in uncovering the origins of adversarial examples.

## ACKNOWLEDGEMENT

This work is supported by Air Force Research Laboratory FA8750-18-2-0058, and U.S. Office of Naval Research. Sijia Liu, Pin-Yu Chen, Huan Zhang and Quanfu Fan were supported by the MIT-IBM Watson Ai Lab, IBM Research.

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

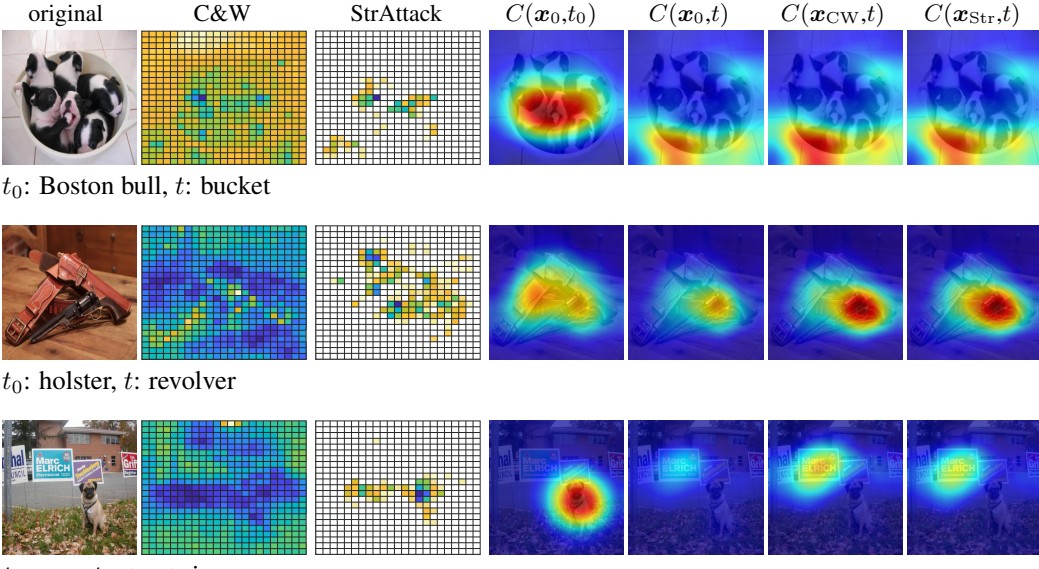

| original | C&W | StrAttack | $C(\boldsymbol{x}_0,t_0)$ | $C(\boldsymbol{x}_0,t)$ | $C(\boldsymbol{x}_{\mathrm{CW}},t)$ | $C(\boldsymbol{x}_{\mathrm{Str}},t)$ |

$t_0$: Boston bull, $t$: bucket

$t_0$: holster, $t$: revolver

$t_0$: pug, $t$: street sign

**Figure 4:** CAMs of adversarial examples generated by the C&W attack and StrAttack under the original label ($t_0$) and target label ($t$). Here $\boldsymbol{x}_{\mathrm{CW}}$ and $\boldsymbol{x}_{\mathrm{Str}}$ denote adversarial examples crafted by different attacks. At each row, the subplots from left to right represent the original image $\boldsymbol{x}_0$, perturbations generated by C&W attack, perturbations generated by StrAttack, and CAMs with respect to natural or adversarial example under $t_0$ or $t$. Here the class $c$ specified CAM with respect to image $\mathbf{x}$ is denoted by $C(\mathbf{x}, c)$.

Tom B Brown, Dandelion Mané, Aurko Roy, Martín Abadi, and Justin Gilmer. Adversarial patch. *arXiv preprint arXiv:1712.09665*, 2017.

N. Carlini and D. Wagner. Towards evaluating the robustness of neural networks. In *Security and Privacy (SP), 2017 IEEE Symposium on*, pp. 39–57. IEEE, 2017.

J. Chen, L. Yang, Y. Zhang, M. Alber, and D. Chen. Combining fully convolutional and recurrent neural networks for 3d biomedical image segmentation. In *Advances in Neural Information Processing Systems*, pp. 3036–3044, 2016.

P.-Y. Chen, Y. Sharma, H. Zhang, J. Yi, and C.-J. Hsieh. Ead: elastic-net attacks to deep neural networks via adversarial examples. *arXiv preprint arXiv:1709.04114*, 2017a.

P.-Y. Chen, H. Zhang, Y. Sharma, J. Yi, and C.-J. Hsieh. Zoo: Zeroth order optimization based black-box attacks to deep neural networks without training substitute models. In *Proceedings of the 10th ACM Workshop on Artificial Intelligence and Security*, pp. 15–26. ACM, 2017b.

J. Deng, W. Dong, R. Socher, L. Li, K. Li, and F.-F. Li. Imagenet: A large-scale hierarchical image database. In *Computer Vision and Pattern Recognition, 2009. CVPR 2009. IEEE Conference on*, pp. 248–255. IEEE, 2009.

Y. Dong, H. Su, J. Zhu, and F. Bao. Towards interpretable deep neural networks by leveraging adversarial examples. *arXiv preprint arXiv:1708.05493*, 2017.

I. Evtimov, K. Eykholt, E. Fernandes, T. Kohno, B. Li, A. Prakash, A. Rahmati, and D. Song. Robust physical-world attacks on machine learning models. *arXiv preprint arXiv:1707.08945*, 2017.

J. Fu, J. Co-Reyes, and S. Levine. Ex2: Exploration with exemplar models for deep reinforcement learning. In *Advances in Neural Information Processing Systems*, pp. 2574–2584, 2017.

Y. Geifman and R. El-Yaniv. Selective classification for deep neural networks. In *Advances in neural information processing systems*, pp. 4885–4894, 2017.

J. Gilmer, L. Metz, F. Faghri, S. Schoenholz, M. Raghu, M. Wattenberg, and I. Goodfellow. Adversarial spheres. *arXiv preprint arXiv:1801.02774*, 2018.

I. Goodfellow, J. Shlens, and C. Szegedy. Explaining and harnessing adversarial examples. *arXiv preprint arXiv:1412.6572*, 2014.

D. Harwath, A. Torralba, and J. Glass. Unsupervised learning of spoken language with visual context. In *Advances in Neural Information Processing Systems*, pp. 1858–1866, 2016.

K. He, X. Zhang, S. Ren, and J. Sun. Deep residual learning for image recognition. In *Proceedings of the IEEE conference on computer vision and pattern recognition*, pp. 770–778, 2016.

G. Hinton, L. Deng, D. Yu, G. Dahl, A.-r. Mohamed, N. Jaitly, A. Senior, V. Vanhoucke, P. Nguyen, T. Sainath, et al. Deep neural networks for acoustic modeling in speech recognition: The shared views of four research groups. *IEEE Signal Processing Magazine*, 29(6):82–97, 2012.

Gao Huang, Zhuang Liu, Laurens Van Der Maaten, and Kilian Q Weinberger. Densely connected convolutional networks. In *CVPR*, volume 1, pp. 3, 2017.

D. Karmon, D. Zoran, and Y. Goldberg. Lavan: Localized and visible adversarial noise. *arXiv preprint arXiv:1801.02608*, 2018.

D. Kingma and J. Ba. Adam: A method for stochastic optimization. *2015 ICLR*, arXiv preprint arXiv:1412.6980, 2015. URL `http://arxiv.org/abs/1412.6980`.

A. Krizhevsky and G. Hinton. Learning multiple layers of features from tiny images. *Master's thesis, Department of Computer Science, University of Toronto*, 2009.

A. Kurakin, I. Goodfellow, and S. Bengio. Adversarial examples in the physical world. *arXiv preprint arXiv:1607.02533*, 2016.

A. Kurakin, I. Goodfellow, and S. Bengio. Adversarial machine learning at scale. *2017 ICLR*, arXiv preprint arXiv:1611.01236, 2017. URL `http://arxiv.org/abs/1611.01236`.

Y. Lecun, L. Bottou, Y. Bengio, and P. Haffner. Gradient-based learning applied to document recognition. *Proceedings of the IEEE*, 86(11):2278–2324, Nov 1998. ISSN 0018-9219. doi: 10.1109/5.726791.

Q. Liu, X. Shen, and Y. Gu. Linearized admm for non-convex non-smooth optimization with convergence analysis. *arXiv preprint arXiv:1705.02502*, 2017.

S. Liu, S. Kar, M. Fardad, and P. K. Varshney. Sparsity-aware sensor collaboration for linear coherent estimation. *IEEE Transactions on Signal Processing*, 63(10):2582–2596, 2015.

S. Liu, J. Chen, P.-Y. Chen, and A. O. Hero. Zeroth-order online admm: Convergence analysis and applications. In *Proceedings of the Twenty-First International Conference on Artificial Intelligence and Statistics*, volume 84, pp. 288–297, April 2018.

A. Madry, A. Makelov, L. Schmidt, D. Tsipras, and A. Vladu. Towards deep learning models resistant to adversarial attacks. *arXiv preprint arXiv:1706.06083*, 2017.

A. Nguyen, J. Yosinski, and J. Clune. Deep neural networks are easily fooled: High confidence predictions for unrecognizable images. In *Proceedings of the IEEE Conference on Computer Vision and Pattern Recognition*, pp. 427–436, 2015.

H. Ouyang, N. He, L. Tran, and A. Gray. Stochastic alternating direction method of multipliers. In *International Conference on Machine Learning*, pp. 80–88, 2013.

N. Papernot, I. Goodfellow, R. Sheatsley, R. Feinman, and P. McDaniel. cleverhans v1.0.0: an adversarial machine learning library. *arXiv preprint arXiv:1610.00768*, 2016a.

N. Papernot, P. McDaniel, S. Jha, M. Fredrikson, B. Celik, and A. Swami. The limitations of deep learning in adversarial settings. In *Security and Privacy (EuroS&P), 2016 IEEE European Symposium on*, pp. 372–387. IEEE, 2016b.

N. Papernot, P. McDaniel, X. Wu, S. Jha, and A. Swami. Distillation as a defense to adversarial perturbations against deep neural networks. In *Security and Privacy (SP), 2016 IEEE Symposium on*, pp. 582–597. IEEE, 2016c.

N. Papernot, P. McDaniel, I. Goodfellow, S. Jha, Z. Celik, and A. Swami. Practical black-box attacks against machine learning. In *Proceedings of the 2017 ACM on Asia Conference on Computer and Communications Security*, pp. 506–519. ACM, 2017.

N. Parikh, S. Boyd, et al. Proximal algorithms. *Foundations and Trends® in Optimization*, 1(3): 127–239, 2014.

M. Sharif, L. Bauer, and M. K. Reiter. On the suitability of $l_p$-norms for creating and preventing adversarial examples. *CoRR*, abs/1802.09653, 2018.

X. Shi, M. Sapkota, F. Xing, F. Liu, L. Cui, and L. Yang. Pairwise based deep ranking hashing for histopathology image classification and retrieval. *Pattern Recognition*, 81:14 – 22, 2018. ISSN 0031-3203. doi: https://doi.org/10.1016/j.patcog.2018.03.015. URL http://www.sciencedirect.com/science/article/pii/S0031320318301055.

D. Silver, A. Huang, C. Maddison, A. Guez, L. Sifre, G. Van Den Driessche, J. Schrittwieser, I. Antonoglou, V. Panneershelvam, M. Lanctot, et al. Mastering the game of go with deep neural networks and tree search. *nature*, 529(7587):484–489, 2016.

A. Sinha, H. Namkoong, and J. Duchi. Certifying some distributional robustness with principled adversarial training. 2018.

Dong Su, Huan Zhang, Hongge Chen, Jinfeng Yi, Pin-Yu Chen, and Yupeng Gao. Is robustness the cost of accuracy?–a comprehensive study on the robustness of 18 deep image classification models. *arXiv preprint arXiv:1808.01688*, 2018.

J. Su, D. Vargas, and S. Kouichi. One pixel attack for fooling deep neural networks. *arXiv preprint arXiv:1710.08864*, 2017.

T. Suzuki. Dual averaging and proximal gradient descent for online alternating direction multiplier method. In *International Conference on Machine Learning*, pp. 392–400, 2013.

C. Szegedy, W. Zaremba, I. Sutskever, J. Bruna, D. Erhan, I. Goodfellow, and R. Fergus. Intriguing properties of neural networks. *arXiv preprint arXiv:1312.6199*, 2013.

C. Szegedy, V. Vanhoucke, S. Ioffe, J. Shlens, and Z. Wojna. Rethinking the inception architecture for computer vision. *2016 IEEE Conference on Computer Vision and Pattern Recognition (CVPR)*, pp. 2818–2826, 2016.

Christian Szegedy, Sergey Ioffe, Vincent Vanhoucke, and Alexander A Alemi. Inception-v4, inception-resnet and the impact of residual connections on learning. In *AAAI*, volume 4, pp. 12, 2017.

F. Tramèr, A. Kurakin, N. Papernot, I. Goodfellow, D. Boneh, and P. McDaniel. Ensemble adversarial training: Attacks and defenses. *2018 ICLR*, arXiv preprint arXiv:1705.07204, 2018.

C. Xiao, J. Zhu, B. Li, W. He, M. Liu, and D. Song. Spatially transformed adversarial examples. *CoRR*, abs/1801.02612, 2018. URL http://arxiv.org/abs/1801.02612.

M. Yuan and Y. Lin. Model selection and estimation in regression with grouped variables. *Journal of the Royal Statistical Society: Series B (Statistical Methodology)*, 68(1):49–67, 2006.

B. Zhou, A. Khosla, A. Lapedriza, A. Oliva, and A. Torralba. Learning deep features for discriminative localization. In *Proceedings of the IEEE Conference on Computer Vision and Pattern Recognition*, pp. 2921–2929, 2016.

## APPENDIX

## A  ILLUSTRATIVE EXAMPLE OF GROUP SPARSITY

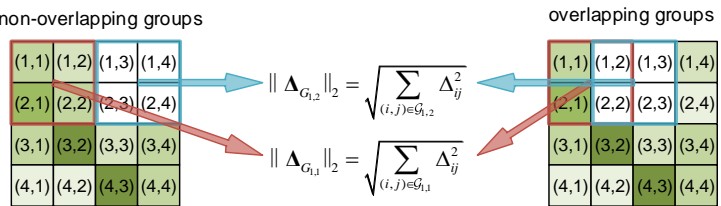

**Figure A1:** An example of $4 \times 4$ perturbation matrix under sliding masks with different strides. The values of matrix elements are represented by color's intensity (white stands for 0). Left: Non-overlapping groups with $r = 2$ and $S = 2$. Right: Overlapping groups with $r = 2$ and $S = 1$. In both cases, two groups $\mathcal{G}_{1,1}$ and $\mathcal{G}_{1,2}$ are highlighted, where $\mathcal{G}_{1,1}$ is non-sparse, and $\mathcal{G}_{1,2}$ is sparse.

## B  PROOF OF PROPOSITION 1

We recall that the augmented Lagrangian function $L(\boldsymbol{\delta}, \mathbf{z}, \mathbf{w}, \mathbf{y}, \mathbf{u}, \mathbf{v}, \mathbf{s})$ is given by

$$
\begin{aligned}
L(\mathbf{z}, \boldsymbol{\delta}, \mathbf{y}, \mathbf{w}, \mathbf{u}, \mathbf{v}, \mathbf{s}) =& f(\mathbf{z} + \mathbf{x}_0) + \gamma D(\boldsymbol{\delta}) + \tau \sum_{i=1}^{PQ} \|\mathbf{y}_{\mathcal{D}_i}\|_2 + h(\mathbf{w}) + \mathbf{u}^T(\boldsymbol{\delta} - \mathbf{z}) \\
& + \mathbf{v}^T(\mathbf{y} - \mathbf{z}) + \mathbf{s}^T(\mathbf{w} - \mathbf{z}) + \frac{\rho}{2}\|\boldsymbol{\delta} - \mathbf{z}\|_2^2 + \frac{\rho}{2}\|\mathbf{y} - \mathbf{z}\|_2^2 + \frac{\rho}{2}\|\mathbf{w} - \mathbf{z}\|_2^2.
\end{aligned} \quad (24)
$$

Problem (7), to minimize $L(\boldsymbol{\delta}, \mathbf{z}^k, \mathbf{w}, \mathbf{y}, \mathbf{u}^k, \mathbf{v}^k, \mathbf{s}^k)$, can be decomposed into three sub-problems:

$$
\underset{\boldsymbol{\delta}}{\text{minimize}} \ \gamma D(\boldsymbol{\delta}) + \frac{\rho}{2}\|\boldsymbol{\delta} - \mathbf{a}\|_2^2, \quad (25)
$$

$$
\underset{\mathbf{w}}{\text{minimize}} \ h(\mathbf{w}) + \frac{\rho}{2}\|\mathbf{w} - \mathbf{b}\|_2^2, \quad (26)
$$

$$
\underset{\mathbf{y}}{\text{minimize}} \ \tau \sum_{i=1}^{PQ} \|\mathbf{y}_{\mathcal{D}_i}\|_2 + \frac{\rho}{2}\|\mathbf{y} - \mathbf{c}\|_2^2, \quad (27)
$$

where $\mathbf{a} := \mathbf{z}^k - \mathbf{u}^k/\rho$, $\mathbf{b} := \mathbf{z}^k - \mathbf{s}^k/\rho$, and $\mathbf{c} := \mathbf{z}^k - \mathbf{v}^k/\rho$.

**$\boldsymbol{\delta}$-step**  Suppose $D(\boldsymbol{\delta}) = \|\boldsymbol{\delta}\|_2^2$, then the solution to problem (25) is easily acquired as below

$$
\boldsymbol{\delta}^{k+1} = \frac{\rho}{\rho + 2\gamma}\mathbf{a} \quad (28)
$$

**w-step**  Based on the definition of $h(\mathbf{w})$, problem (26) becomes

$$
\begin{aligned}
&\underset{\mathbf{w}}{\text{minimize}} \quad \|\mathbf{w} - \mathbf{b}\|_2^2 \\
&\text{subject to} \quad (\mathbf{x}_0 + \mathbf{w}) \in [0, 1]^n, \ \|\mathbf{w}\|_\infty \le \epsilon.
\end{aligned} \quad (29)
$$

Problem (29) is equivalent to

$$
\begin{aligned}
&\underset{w_i}{\text{minimize}} \quad (w_i - a_i)_2^2 \\
&\text{subject to} \quad -[\mathbf{x}_0]_i \le w_i \le 1 - [\mathbf{x}_0]_i, \ |w_i| \le \epsilon
\end{aligned} \quad (30)
$$

for $i \in [n]$, where $x_i$ or $[\mathbf{x}]_i$ represents the $i$th element of $\mathbf{x}$, and $1 - [\mathbf{x}_0]_i > 0$ since $[\mathbf{x}_0]_i \in [0, 1]$. Problem (30) then yields the solution

$$
[\mathbf{w}^{k+1}]_i = \begin{cases} \min\{1 - [\mathbf{x}_0]_i, \epsilon\} & a_i > \min\{1 - [\mathbf{x}_0]_i, \epsilon\} \\ \max\{-[\mathbf{x}_0]_i, -\epsilon\} & a_i < \max\{-[\mathbf{x}_0]_i, -\epsilon\} \\ a_i & \text{otherwise.} \end{cases} \quad (31)
$$

**y-step** Problem (27) becomes

$$\underset{\mathbf{y}}{\text{minimize}} \quad \sum_{i=1}^{PQ} \|\mathbf{y}_{\mathcal{D}_i}\|_2 + \frac{\rho}{2\tau}\|\mathbf{y} - \mathbf{c}\|_2^2, \tag{32}$$

The solution is given by the proximal operator associated with the $\ell_2$ norm with parameter $\tau/\rho$ (Parikh et al., 2014)

$$[\mathbf{y}^{k+1}]_{\mathcal{D}_i} = \left(1 - \frac{\tau}{\rho\|[\mathbf{c}]_{\mathcal{D}_i}\|_2}\right)_+ [\mathbf{c}]_{\mathcal{D}_i}, \ i \in [PQ], \tag{33}$$

where recall that $\cup_{i\in[PQ]}\mathcal{D}_i = [n]$, and $\mathcal{D}_i \cap \mathcal{D}_j = \emptyset$ if $i \neq j$. $\qquad\square$

## C  PROOF OF PROPOSITION 2

The augmented Lagrangian of problem (16) is given by

$$L(\mathbf{z}, \boldsymbol{\delta}, \mathbf{w}, \{\mathbf{y}_i\}, \mathbf{u}, \mathbf{v}_i, \mathbf{s}) = f(\mathbf{z} + \mathbf{x}_0) + \gamma D(\boldsymbol{\delta}) + h(\mathbf{w}) + \tau \sum_{i=1}^{PQ} \|\mathbf{y}_{i,\mathcal{D}_i}\|_2 + \mathbf{u}^T(\boldsymbol{\delta} - \mathbf{z}) + \mathbf{s}^T(\mathbf{w} - \mathbf{z})$$

$$+ \sum_{i=1}^{PQ} \mathbf{v}_i^T(\mathbf{y}_i - \mathbf{z}) + \frac{\rho}{2}\|\boldsymbol{\delta} - \mathbf{z}\|_2^2 + \frac{\rho}{2}\|\mathbf{w} - \mathbf{z}\|_2^2 + \frac{\rho}{2}\sum_{i=1}^{PQ}\|\mathbf{y}_i - \mathbf{z}\|_2^2, \tag{34}$$

where $\mathbf{u}$, $\mathbf{v}_i$ and $\mathbf{s}$ are the Lagrangian multipliers.

ADMM decomposes the optimization variables into *two* blocks and adopts the following iterative scheme,

$$\{\boldsymbol{\delta}^{k+1}, \mathbf{w}^{k+1}, \mathbf{y}_i^{k+1}\} = \underset{\boldsymbol{\delta},\mathbf{w},\{\mathbf{y}_i\}}{\arg\min} L(\mathbf{z}^k, \boldsymbol{\delta}, \mathbf{w}, \mathbf{y}_i, \mathbf{u}^k, \mathbf{v}_i^k, \mathbf{s}^k), \tag{35}$$

$$\mathbf{z}^{k+1} = \underset{\mathbf{z}}{\arg\min} L(\mathbf{z}, \boldsymbol{\delta}^{k+1}, \mathbf{w}^{k+1}, \mathbf{y}_i^{k+1}, \mathbf{u}^k, \mathbf{v}_i^k, \mathbf{s}^k), \tag{36}$$

$$\begin{cases} \mathbf{u}^{k+1} = \mathbf{u}^k + \rho(\boldsymbol{\delta}^{k+1} - \mathbf{z}^{k+1}), \\ \mathbf{v}_i^{k+1} = \mathbf{v}_i^k + \rho(\mathbf{y}_i^{k+1} - \mathbf{z}^{k+1}), \text{ for } i \in [PQ], \\ \mathbf{s}^{k+1} = \mathbf{s}^k + \rho(\mathbf{w}^{k+1} - \mathbf{z}^{k+1}), \end{cases} \tag{37}$$

where $k$ is the iteration index. Problem (35) can be split into three subproblems as shown below,

$$\underset{\boldsymbol{\delta}}{\text{minimize}} \ \gamma D(\boldsymbol{\delta}) + \frac{\rho}{2}\|\boldsymbol{\delta} - \mathbf{a}\|_2^2, \tag{38}$$

$$\underset{\mathbf{w}}{\text{minimize}} \ h(\mathbf{w}) + \frac{\rho}{2}\|\mathbf{w} - \mathbf{b}\|_2^2, \tag{39}$$

$$\underset{\mathbf{y}_i}{\text{minimize}} \ \tau\|\mathbf{y}_{i,\mathcal{D}_i}\|_2 + \frac{\rho}{2}\|\mathbf{y}_i - \mathbf{c}_i\|_2^2, \text{ for } i \in [PQ]. \tag{40}$$

where $\mathbf{a} = \mathbf{z}^k - \mathbf{u}^k/\rho$, $\mathbf{b} = \mathbf{z}^k - \mathbf{s}^k/\rho$ and $\mathbf{c}_i = \mathbf{z}^k - \mathbf{v}_i^k/\rho$. Each problem has a closed form solution. Note that the solutions to problem (38) and problem (39) are given (28) and (31).

**$\mathbf{y}_i$-step** Problem (40) can be rewritten as

$$\underset{\mathbf{y}_i}{\text{minimize}} \ \tau\|\mathbf{y}_{i,\mathcal{D}_i}\|_2 + \frac{\rho}{2}\|\mathbf{y}_{i,\mathcal{D}_i} - [\mathbf{c}_i]_{\mathcal{D}_i}\|_2^2 + \frac{\rho}{2}\|\mathbf{y}_{i,[n]/\mathcal{D}_i} - [\mathbf{c}_i]_{[n]/\mathcal{D}_i}\|_2^2, \text{ for } i \in [PQ], \tag{41}$$

which can be decomposed into

$$\underset{\mathbf{y}_{i,\mathcal{D}_i}}{\text{minimize}} \ \tau\|\mathbf{y}_{i,\mathcal{D}_i}\|_2 + \frac{\rho}{2}\|\mathbf{y}_{i,\mathcal{D}_i} - [\mathbf{c}_i]_{\mathcal{D}_i}\|_2^2, \text{ for } i \in [PQ], \tag{42}$$

and

$$\underset{\mathbf{y}_{i,[n]/\mathcal{D}_i}}{\text{minimize}} \; \|\mathbf{y}_{i,[n]/\mathcal{D}_i} - [\mathbf{c}_i]_{[n]/\mathcal{D}_i}\|_2^2, \text{ for } i \in [PQ]. \tag{43}$$

The solution to problem (42) can be obtained through the block soft thresholding operator (Parikh et al., 2014),

$$\left[\mathbf{y}_i^{k+1}\right]_{\mathcal{D}_i} = \left(1 - \frac{\tau}{\rho\|[\mathbf{c}_i]_{\mathcal{D}_i}\|_2}\right)_+ [\mathbf{c}_i]_{\mathcal{D}_i}, \text{ for } i \in [PQ], \tag{44}$$

The solution to problem (43) is given by,

$$\left[\mathbf{y}_i^{k+1}\right]_{[n]/\mathcal{D}_i} = [\mathbf{c}_i]_{[n]/\mathcal{D}_i}, \text{ for } i \in [PQ]. \tag{45}$$

**z-step**  Problem (36) can be simplified to

$$\underset{\mathbf{z}}{\text{minimize}} \quad f(\mathbf{x}_0 + \mathbf{z}) + \frac{\rho}{2}\|\mathbf{z} - \mathbf{a}'\|_2^2 + \frac{\rho}{2}\|\mathbf{z} - \mathbf{b}'\|_2^2 + \frac{\rho}{2}\sum_{i=1}^{PQ}\|\mathbf{z} - \mathbf{c}_i'\|_2^2, \tag{46}$$

where $\mathbf{a}' := \boldsymbol{\delta}^{k+1} + \mathbf{u}^k/\rho$, $\mathbf{b}' := \mathbf{w}^{k+1} + \mathbf{s}^k/\rho$, and $\mathbf{c}_i' := \mathbf{y}_i^{k+1} + \mathbf{v}_i^k/\rho$. We solve problem (46) using the linearization technique (Suzuki, 2013; Liu et al., 2018; Boyd et al., 2011). More specifically, the function $f$ is replaced with its first-order Taylor expansion at the point $\mathbf{z}^k$ by adding a Bregman divergence term $(\eta_k/2)\|\mathbf{z} - \mathbf{z}^k\|_2^2$. As a result, problem (46) becomes

$$\underset{\mathbf{z}}{\text{minimize}} \quad (\nabla f(\mathbf{z}^k + \mathbf{x}_0))^T(\mathbf{z} - \mathbf{z}^k) + \frac{\eta_k}{2}\|\mathbf{z} - \mathbf{z}^k\|_2^2 + \frac{\rho}{2}\|\mathbf{z} - \mathbf{a}'\|_2^2$$
$$+ \frac{\rho}{2}\|\mathbf{z} - \mathbf{b}'\|_2^2 + \frac{\rho}{2}\sum_{i=1}^{PQ}\|\mathbf{z} - \mathbf{c}_i'\|_2^2, \tag{47}$$

whose solution is given by

$$\mathbf{z}^{k+1} = \frac{\eta_k\mathbf{z}^k + \rho\mathbf{a}' + \rho\mathbf{b}' + \rho\sum_{i=1}^{PQ}\mathbf{c}_i' - \nabla f(\mathbf{z}^k + \mathbf{x}_0)}{\eta_k + (2 + PQ)\rho}. \tag{48}$$

$\square$

# D  PROOF OF PROPOSITION 3

We start by converting problem (20) into the ADMM form

$$\underset{\boldsymbol{\delta},\mathbf{z}}{\text{minimize}} \quad f(\mathbf{x}_0 + \mathbf{z}) + g(\mathbf{z}) + \gamma D(\boldsymbol{\delta}) + h(\boldsymbol{\delta}) + g(\boldsymbol{\delta})$$
$$\text{subject to} \quad \boldsymbol{\delta} = \mathbf{z}, \tag{49}$$

where $\mathbf{z}$ and $\boldsymbol{\delta}$ are optimization variables, $g(\delta)$ is an indicator function with respect to the constraint $\{\delta_i = 0, \text{ if } i \in \mathcal{S}_\sigma\}$, and $h(\boldsymbol{\delta})$ is the other indicator function with respect to the other constraints $(\mathbf{x}_0 + \boldsymbol{\delta}) \in [0,1]^n$, $\|\boldsymbol{\delta}\|_\infty \leq \epsilon$.

The augmented Lagrangian of problem (20) is given by

$$L(\boldsymbol{\delta}, \mathbf{z}, \mathbf{u}) = f(\mathbf{z} + \mathbf{x}_0) + g(\mathbf{z}) + \gamma D(\boldsymbol{\delta}) + h(\boldsymbol{\delta}) + g(\boldsymbol{\delta}) + \mathbf{u}^T(\boldsymbol{\delta} - \mathbf{z}) + \frac{\rho}{2}\|\boldsymbol{\delta} - \mathbf{z}\|_2^2, \tag{50}$$

where $\mathbf{u}$ is the Lagrangian multiplier.

ADMM yields the following alternating steps

$$\boldsymbol{\delta}^{k+1} = \underset{\boldsymbol{\delta}}{\arg\min}\, L(\boldsymbol{\delta}, \mathbf{z}^k, \mathbf{u}^k) \tag{51}$$

$$\mathbf{z}^{k+1} = \underset{\mathbf{z}}{\arg\min}\, L(\boldsymbol{\delta}^{k+1}, \mathbf{z}, \mathbf{u}^k) \tag{52}$$

$$\mathbf{u}^{k+1} = \mathbf{u}^k + \rho(\boldsymbol{\delta}^{k+1} - \mathbf{z}^{k+1}). \tag{53}$$

**$\delta$-step**  Suppose $D(\delta) = \|\delta\|_2^2$, problem (51) becomes

$$
\begin{aligned}
\underset{\delta}{\text{minimize}} \quad & \gamma\|\delta\|_2^2 + \tfrac{\rho}{2}\|\delta - \mathbf{a}\|_2^2 \\
\text{subject to} \quad & (\mathbf{x}_0 + \delta) \in [0,1]^n, \ \|\delta\|_\infty \le \epsilon \\
& \delta_i = 0, \ \text{if } i \in \mathcal{S}_\sigma,
\end{aligned}
\tag{54}
$$

where $\mathbf{a} := \mathbf{z}^k - \mathbf{u}^k/\rho$. Problem (54) can be decomposed elementwise

$$
\begin{aligned}
\underset{\delta_i}{\text{minimize}} \quad & \tfrac{2\gamma+\rho}{\rho}\delta_i^2 - 2a_i\delta_i + a_i^2 = \tfrac{2\gamma+\rho}{\rho}\left(\delta_i - \tfrac{\rho}{2\gamma+\rho}a_i\right)^2 \\
\text{subject to} \quad & ([\mathbf{x}_0]_i + \delta_i) \in [0,1], \ |\delta_i| \le \epsilon \\
& \delta_i = 0, \ \text{if } i \in \mathcal{S}_\sigma.
\end{aligned}
\tag{55}
$$

The solution to problem (55) is then given by

$$
[\delta^{k+1}]_i = \begin{cases}
0 & i \in \mathcal{S}_\sigma \\
\min\{1 - [\mathbf{x}_0]_i, \epsilon\} & \tfrac{\rho}{2\gamma+\rho}a_i > \min\{1 - [\mathbf{x}_0]_i, \epsilon\}, \ i \notin \mathcal{S}_\sigma \\
\max\{-[\mathbf{x}_0]_i, -\epsilon\} & \tfrac{\rho}{2\gamma+\rho}a_i < \max\{-[\mathbf{x}_0]_i, -\epsilon\}, i \notin \mathcal{S}_\sigma \\
\tfrac{\rho}{2\gamma+\rho}a_i & \text{otherwise.}
\end{cases}
\tag{56}
$$

**z-step**  Problem (52) yields

$$
\begin{aligned}
\underset{\mathbf{z}}{\text{minimize}} \quad & f(\mathbf{x}_0 + \mathbf{z}) + \frac{\rho}{2}\|\mathbf{z} - \mathbf{a}'\|_2^2 \\
\text{subject to} \quad & z_i = 0, \ \text{if } i \in \mathcal{S}_\sigma,
\end{aligned}
\tag{57}
$$

where $\mathbf{a}' = \delta^{k+1} + \mathbf{u}^k/\rho$. We solve problem (57) using the linearization technique (Suzuki, 2013; Liu et al., 2018; Boyd et al., 2011),

$$
\begin{aligned}
\underset{\mathbf{z}}{\text{minimize}} \quad & (\nabla f(\mathbf{x}_0 + \mathbf{z}^k))^T(\mathbf{z} - \mathbf{z}^k) + \frac{\eta_k}{2}\|\mathbf{z} - \mathbf{z}^k\|_2^2 + \frac{\rho}{2}\|\mathbf{z} - \mathbf{a}'\|_2^2 \\
\text{subject to} \quad & z_i = 0, \ \text{if } i \in \mathcal{S}_\sigma,
\end{aligned}
\tag{58}
$$

where $\eta_k$ is a decaying parameter associated with the Bregman divergence term $\|\mathbf{z} - \mathbf{z}^k\|_2^2$. In problems (57) and (58), only variables $\{z_i\}$ satisfying $i \notin \mathcal{S}_\sigma$ are unknown. The solution to problem (58) is then given by

$$
[\mathbf{z}^{k+1}]_i = \begin{cases}
0 & i \in \mathcal{S}_\sigma \\
\frac{\eta_k[\mathbf{z}^k]_i + \rho[\mathbf{a}']_i - [\nabla f(\mathbf{z}^k + \mathbf{x}_0)]_i}{\eta_k + \rho} & i \notin \mathcal{S}_\sigma.
\end{cases}
\tag{59}
$$

$\square$

# E  ADVERSARIAL SALIENCY MAP (ASM) AND CLASS ACTIVATION MAPPING (CAM)

$\mathrm{ASM}(\mathbf{x}, t) \in \mathbb{R}^d$ is defined by the forward derivative of a neural network given the input sample $\mathbf{x}$ and the target label $t$ (Papernot et al., 2016b)

$$
\mathrm{ASM}(\mathbf{x}, t)[i] = \begin{cases}
0 & \text{if } \frac{\partial Z(\mathbf{x})_t}{\partial \mathbf{x}_i} < 0 \text{ or } \sum_{j \ne t}\frac{\partial Z(\mathbf{x})_j}{\partial \mathbf{x}_i} > 0 \\
\left(\frac{\partial Z(\mathbf{x})_t}{\partial \mathbf{x}_i}\right)\left|\sum_{j \ne t}\frac{\partial Z(\mathbf{x})_j}{\partial \mathbf{x}_i}\right| & \text{otherwise,}
\end{cases}
\tag{60}
$$

where $Z(\mathbf{x})_j$ is the $j$th element of logits $Z(\mathbf{x})$, representing the output before the last softmax layer in DNNs. If there exist many classes in a dataset (e.g., 1000 classes in ImageNet), then computing $\sum_{j \ne t}\frac{\partial Z(\mathbf{x})_j}{\partial \mathbf{x}_i}$ is intensive. To circumvent the scalability issue of ASM, we focus on the logit change with respect to the true label $t_0$ and the target label $t$ only. More specifically, we consider three quantities, $\frac{\partial Z(\mathbf{x})_t}{\partial \mathbf{x}_i}$, $-\frac{\partial Z(\mathbf{x})_0}{\partial \mathbf{x}_i}$, and $\left(\frac{\partial Z(\mathbf{x})_t}{\partial \mathbf{x}_i}\right)\left|\sum_{j \ne t}\frac{\partial Z(\mathbf{x})_j}{\partial \mathbf{x}_i}\right|$, which correspond to a) promotion of the score of the target label $t$, b) suppression of the classification score of the true label $t_0$, and c) a dual role on suppression and promotion. As a result, we modify (60) as

$$
\mathrm{ASM}(\mathbf{x}, t)[i] = \begin{cases}
0 & \text{if } \frac{\partial Z(\mathbf{x})_t}{\partial \mathbf{x}_i} < 0 \text{ or } \frac{\partial Z(\mathbf{x})_{t_0}}{\partial \mathbf{x}_i} > 0 \\
\left(\frac{\partial Z(\mathbf{x})_t}{\partial \mathbf{x}_i}\right)\left|\frac{\partial Z(\mathbf{x})_{t_0}}{\partial \mathbf{x}_i}\right| & \text{otherwise.}
\end{cases}
\tag{61}
$$

CAM allows us to visualize the perturbation of adversaries on predicted class scores given any pair of image and object label, and highlights the discriminative object regions detected by CNNs (Zhou et al., 2016). In Fig. A2, we show ASM and the discriminative regions identified by CAM on several ImageNet samples.

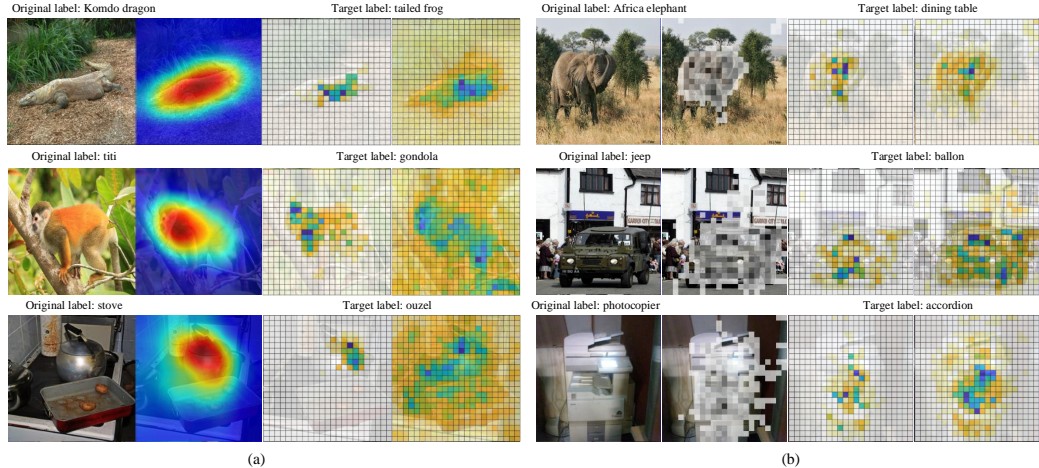

**Figure A2:** (a) Overlay ASM and $\mathbf{B}_{\mathrm{ASM}} \circ \boldsymbol{\delta}$ on top of image with the true and the target label. From left to right: original image, ASM (darker color represents larger value of ASM score), $\mathbf{B}_{\mathrm{ASM}} \circ \boldsymbol{\delta}$ under our attack, and $\mathbf{B}_{\mathrm{ASM}} \circ \boldsymbol{\delta}$ under C&W attack. Here $\nu$ in $\mathbf{B}_{\mathrm{ASM}}$ is set by the 90th percentile of ASM scores. (b) From left to right: original image, CAM of original label, and perturbations with target label generated from the StrAttack and C&W attack, respectively.

## F   EXPERIMENT SETUP AND PARAMETER SETTING

In this work, we consider targeted adversarial attacks since they are believed stronger than untargeted attacks. For targeted attacks, we have different methods to choose the target labels. The average case selects the target label randomly among all the labels that are not the correct label. The best case performs attacks using all incorrect labels, and report the target label that is the least difficult to attack. The worst case performs attacks using all incorrect labels, and report the target label which is the most difficult to attack.

In our experiments, two networks are trained for MNIST and CIFAR-10, respectively, and a pre-trained network is utilized for ImageNet. The model architectures for MNIST and CIFAR-10 are the same, both with four convolutional layers, two max pooling layers, two fully connected layers and a softmax layer. It can achieve 99.5% and 80% accuracy on MNIST and CIFAR-10, respectively. For ImageNet, a pre-trained Inception v3 network (Szegedy et al., 2016) is applied which can achieve 96% top-5 accuracy. All experiments are conducted on machines with NVIDIA GTX 1080 TI GPUs.

The implementations of FGM and IFGM are based on the CleverHans package (Papernot et al., 2016a). The key distortion parameter $\epsilon$ is determined by a fine-grained grid search. For IFGM, we perform 10 FGM iterations and the distortion parameter $\epsilon'$ is set to $\epsilon/10$ for effectiveness as shown in Tramèr et al. (2018). The implementation of the C&W attack is based on the opensource code provided by Carlini & Wagner (2017). The maximum iteration number is set to 1000 and it has 9 binary search steps.

In the StrAttack, the group size for MNIST and CIFAR-10 is $2 \times 2$ and its stride is set to 2 if the non-overlapping mask is used, otherwise the group size is $3 \times 3$ and stride is 2. The group size for ImageNet is $13 \times 13$ and its stride is set to 13. In ADMM, the parameter $\rho$ achieves a trade-off between the convergence rate and the convergence value. A larger $\rho$ could make ADMM converging faster but usually leads to perturbations with larger $\ell_p$ distortion values. A proper configuration of the parameters is suggested as follows: We set the penalty parameter $\rho = 1$, decaying parameter in (14) $\eta_1 = 5$, $\tau = 2$ and $\gamma = 1$. Moreover, we set $c$ defined in (3) to 0.5 for MNIST, 0.25 for CIFAR-10, and 2.5 for ImageNet. Refined attack technique proposed in Sec. 4.2 is applied for all

experiments, we set $\sigma$ is equal to 3% quantile value of non-zero perturbation in $\delta^*$. We observe that 73% of $\delta^*$ can be retrained to a $\sigma$-sparse perturbation successfully which proof the effective of our refined attack step.

# G  SUPPLEMENTARY EXPERIMENTAL RESULTS

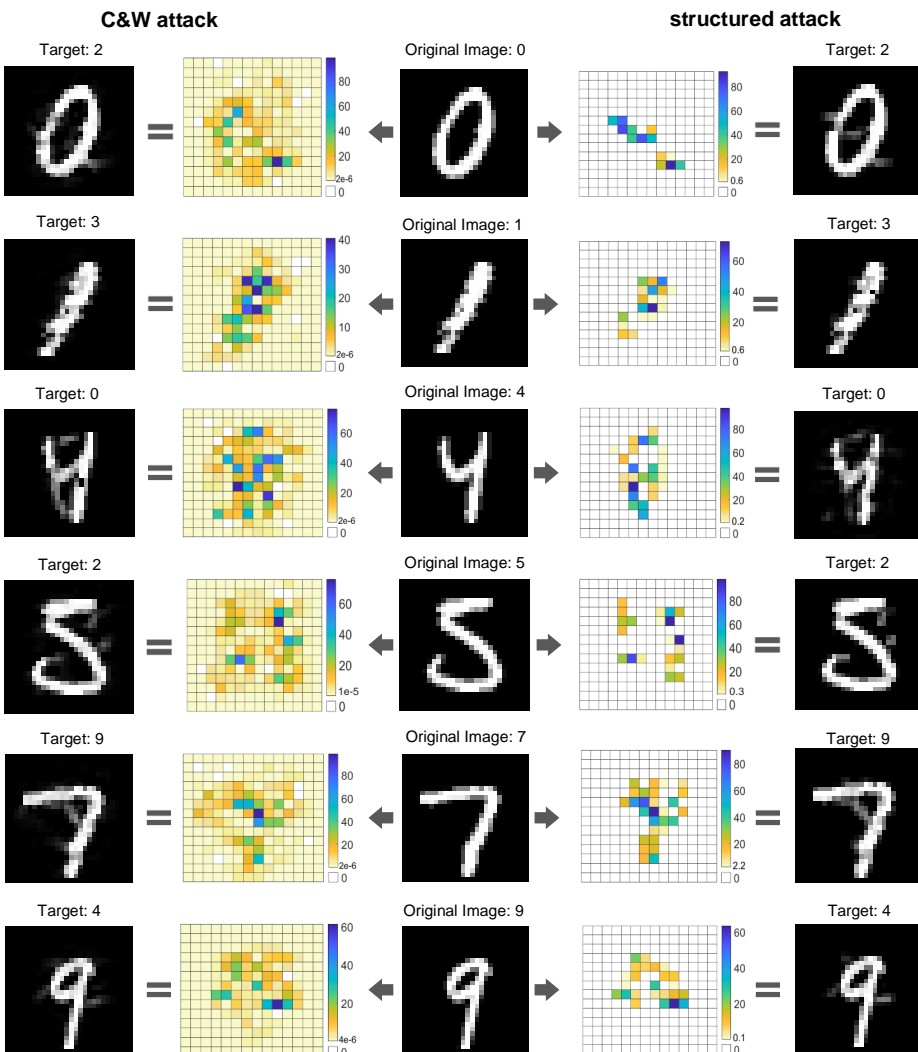

**Figure A3:** C&W attack vs StrAttack on MNIST with grid size $2 \times 2$.

Some random choice samples from MNIST (Fig. A3), CIFAR-10 (Fig. A4) and ImageNet (Fig. A5) compare StrAttack with C&W attack. For better sparse visual effect, we only show non-overlapping mask function results here. From these samples, we can discover a consistent phenomenon that our StrAttack is more interested in some particular regions, they usually appear on the objects or their edges in original images, distinctly seen in MNIST (Fig. A3) and ImageNet (Fig. A5).

# H  STRATTACK AGAINST DEFENSIVE DISTILLATION AND ADVERSARIAL TRAINING

In this section, we present the performance of the StrAttack against defensive distillation (Papernot et al., 2016c) and adversarial training (Tramèr et al., 2018). In defensive distillation, we evaluate the

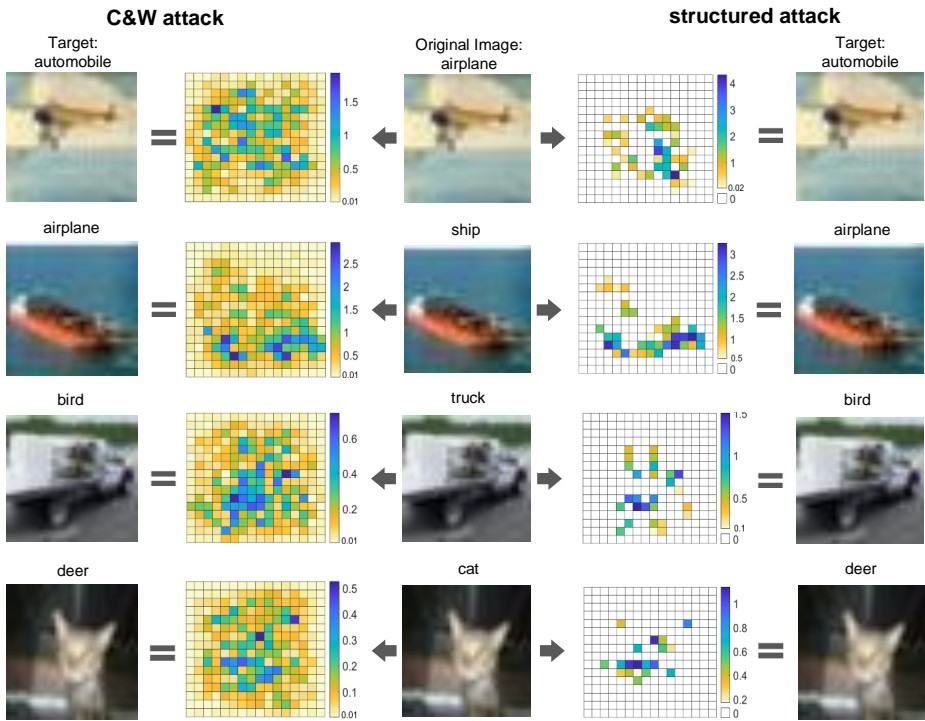

**Figure A4:** C&W attack vs StrAttack on CIFAR-10 with grid size $2 \times 2$.

StrAttack for different temperature parameters on MNIST and CIFAR-10. We generate 9000 adversarial examples with 1000 randomly selected images from MNIST and CIFAR-10, respectively. The attack success rates of the StrAttack for different temperatures $T$ are all 100%. The reason is that distillation at temperature $T$ makes the logits approximately $T$ times larger but does not change the relative values of logits. The StrAttack which works on the relative values of logits does not fail.

We further use the StrAttack to break DNNs training on adversarial examples (Tramèr et al., 2018) with their correct labels on MNIST. The StrAttack is performed on three neural networks: the first network is unprotected, the second is obtained by retraining with 9000 C&W adversarial examples, and the third network is retained with 9000 adversarial examples crafted by the StrAttack. The success rate and distortions on the three networks are shown in Table A1. The StrAttack can break all three networks with 100% success rate. However, adversarial training shows certain defense effects as an increase on the $\ell_1$ or $\ell_2$ distortion on the latter two networks over the unprotected network is observed.

**Table A1:** StrAttack against adversarial training on MNIST

| Adversarial training | Best case | | | Average case | | | Worst case | | |
|---|---|---|---|---|---|---|---|---|---|
| | ASR | $\ell_1$ | $\ell_2$ | ASR | $\ell_1$ | $\ell_2$ | ASR | $\ell_1$ | $\ell_2$ |
| None | 100 | 10.9 | 1.51 | 100 | 18.05 | 2.16 | 100 | 26.9 | 2.81 |
| C&W | 100 | 16.1 | 1.87 | 100 | 25.1 | 2.58 | 100 | 34.2 | 3.26 |
| structured | 100 | 15.6 | 1.86 | 100 | 25.1 | 2.61 | 100 | 34.6 | 3.31 |

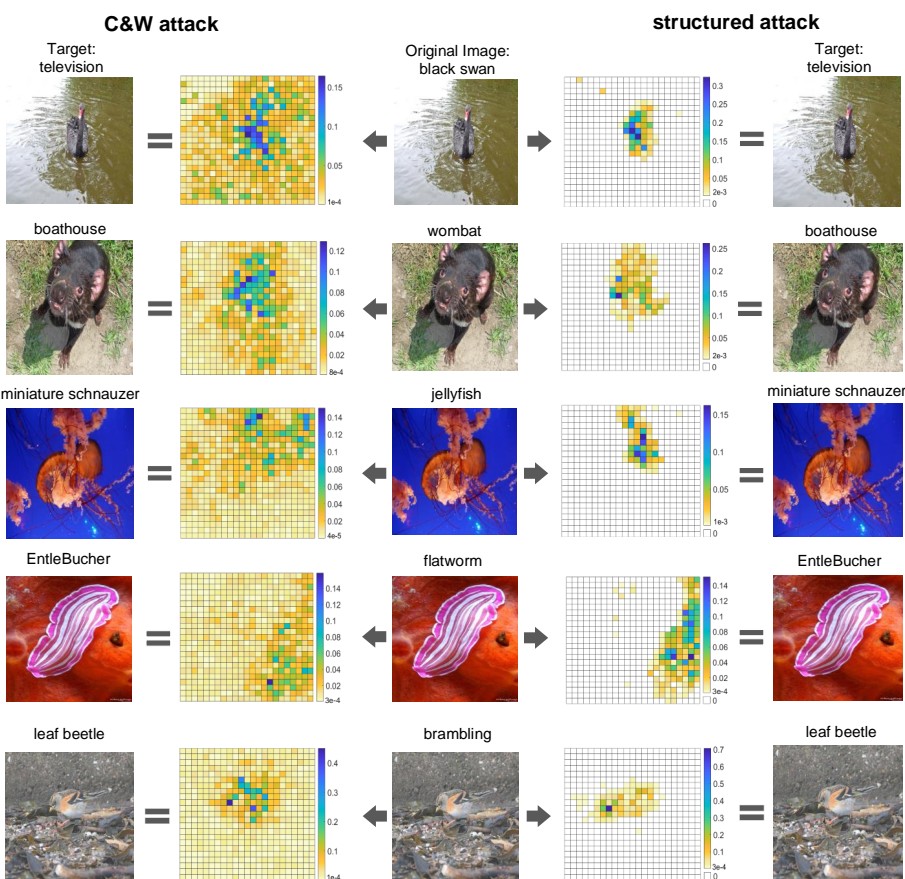

**Figure A5:** C&W attack vs StrAttack on ImageNet with grid size $13 \times 13$.

