# OpenReview forum: "Structured Adversarial Attack:  Towards General Implementation and Better Interpretability"
_ICLR.cc/2019/Conference_

### Official Review · AnonReviewer2 · 2018-11-02
**the paper has good technical qualities, but motivation for the research is not explained**

**Rating:** 6
**Confidence:** 2

**Review:**

The paper proposes a method to find adversarial examples in which the changes are localized to small regions of the image. A group-sparsity objective is introduced for this purpose and it is combined with an l_p objective that was used in prior work to define proximity to the original example. ADMM is applied to maximize the defined objective. It is shown that adversarial examples in which all changes are concentrated in just few regions can be found with the proposed method.

The paper is clearly written and results are convincing. But what I am not sure I understand is what is the purpose of this research. Among the 4 contributions listed in the end of the intro only the last one, Interpretability, seems to have a potential in terms on the impact. Yet am not quite sure how “obtained group-sparse adversarial patterns better shed light on the mechanisms of adversarial perturbations”. I think the mechanisms of adversarial perturbations remain as unclear as they were before this paper.

I am not ready to recommend acceptance of this paper, because I think the due effort to explain the motivation for research and its potential impacts has not been done in this case.

UPD: the discussion and the edits with the authors convinced me that I may have been a bit too strict. I have changed my score from 5 to 6.

---

> ### Author Response · Authors · 2018-11-07
> **Some clarification on our motivation, contributions and potential impacts**
>
> We really thank the reviewer for the insightful comments. As a prompt response, we would like to use this opportunity to reiterate and clarify our motivation, contributions and their potential impacts. Meanwhile, we are also preparing a revision to better address the reviewer's comments.
>
> a) The first contribution "Structure-driven attack" actually indicates the existence of a more stealthy pixel-level adversarial attack under the same norm-bounded threat model, which has not been entirely explored in existing attacks. The motivation of our research stems from devising a more efficient attack that takes advantages of two attacks using extremely opposite principles - C\&W attack (or \ell_infty attacks such as I-FGSM) that modifies all pixels, and one-pixel attack (Su et al., 2017) that only modifies a few pixels. The C\&W attack can achieve small \ell_infty perturbations but has to perturb most pixels (large \ell_0 norm), while the one-pixel attack can achieve extremely small \ell_0 norm but with much higher \ell_infty norm.
>
> Both attack methods may lead to higher noise visibility due to perturbing too many pixels or perturbing a few pixels too much. Motivated by these attack methods and under the same threat model (e.g., \ell_infty constraint), we wonder if there exists a more effective attack that can be as successful as existing attacks but only requires to modify a small subset of pixels. We show that StrAttack is indeed the desired adversarial attack. It is also worth mentioning that one pixel attack has much lower attack success rate on ImageNet than CW and ours.
>
> Consequently, the impacts of StrAttack include (i): understanding why the identified regions in the image are vulnerable to adversarial attacks; and (ii) investigating how the identified attack sparse patterns can benefit adversarial attacks/defenses
>
> b) The second and the third contributions are our technical contributions from the algorithmic perspective. The results indicate that powerful attacks could be derived from more advanced optimization techniques. Note that the proposed StrAttack problem formulation cannot be solved using standard optimization solvers, e.g., Adam, or proximal gradient algorithm, etc, due to the presence of non-smooth regularizers and hard constraints. To address this technique challenge, we proposed the ADMM solution which is quite new for finding adversarial perturbations and enjoys the benefit of having an analytical solution at every ADMM subproblem.
>
> c) We thank R2 for acknowledging interpretability as an impactful contribution. The proposed idea indeed helps researchers to better explain and visualize the effect of adversarial perturbations. Our experimental results, e.g., Figure 1 and 3, clearly show that why we could perturb less but `right` pixels  (with group-sparse patterns) to fool DNNs. Those `right` pixels are the most sensitive pixels to affect the output of classifiers, checked by adversarial saliency analysis in Sec. 6. They also correspond to the most discriminative region of a class activation map, which demonstrates the interpretability of the proposed structured attack. Also, we would like to clarify that "The mechanisms of adversarial perturbations " meant the above findings. Based on the feedback, we now realize that  'mechanisms' might not be the best word to describe our contribution, and thus we will rephrase our claim and make it clearer and more accurate. Note that many adversarial attack methods were proposed in the literature, however, few of them linked interpretability with adversarial examples.

---

> > ### Comment · AnonReviewer2 · 2018-11-10
> > **Thank you for the clarifications**
> >
> > Thank you for the clarifications, in particular for the item (a), that explains better why this research is important. I will take a look at the revision when you upload it and I will consider reevaluating your paper.

---

> > > ### Author Response · Authors · 2018-11-25
> > > **Response to Reviewer2**
> > >
> > > As we discussed earlier, the motivation of our research is to seek a more effective attack, which can be as successful as existing attacks (in terms of achieving the same attack success rate and keeping small L1, L2 and L_infty distortion), but only requires to modify a small subset of pixels. We show that StrAttack is indeed the desired adversarial attack.
> > >
> > >
> > > In the revised paper, we strengthen the potential impacts of StrAttack from the aspects a) performance of attacking robust adversarially trained model, a) attack transferability, and c) interpretability of complex images.
> > >
> > >
> > > First, we show the powerfulness of StrAttack to attack the defensive model obtained from robust adversarial training [Madry et al. 2018], which is commonly regarded as the strongest defense on MNIST. As we can see, although StrAttack perturbs much less pixels, its attack success rate does not drop. This implies that we could perturb less but ‘right’ pixels  (with better interpretable adversarial patterns) without losing its attack performance.
> > >
> > > Second, we compare the transferability of StrAttack to other attacks. Here the transferability is characterized by the attack success rate of adversarial examples (found by one attack generation method against a given network model) transferred to another different network model. We present the transferability of 3 attacks from the model Inception V3 to model Inception V3, Inception V4, ResNet 50, ResNet 152, DenseNet 121 and DenseNet  161. As we can see, StrAttack yields the highest transferability almost at every model. We refer the reviewer to Table 3 for more details.
> > >
> > >
> > > Third, we show more examples to visualize the interpretability of adversarial perturbations on certain complex images. In the ‘pug’-‘street sign’ example of Fig. 4, objects of the original label (pug) and the target label (street sign) exist simultaneously. As we can see, adversarial perturbations generated from StrAttack are perfectly matched to the most discriminative image regions localized by CAM: the adversary shows suppression on the discriminative region of the original label and promotion on the discriminative region of the target label. By contrast, the CW attack is less interpretable due to the high noise visibility (perturbing too many pixels).

---

### Official Review · AnonReviewer1 · 2018-11-05
**An interesting but not entirely novel contribution**

**Rating:** 7
**Confidence:** 3

**Review:**

The paper proposes a novel approach to generate adversarial examples based on structured sparsity principles. In particular the authors focus on the intuition that adversarial examples in computer vision might benefit from encoding information about the local structure of the data. To this end, lp *group* norms can be used in contrast to standard global lp norms when constraining or penalizing the optimization of the adversarial example. The authors propose an optimization strategy to address this problem. The authors evaluate the proposed approach on real data, comparing it against state-of-the-art competitors, which do not leverage the structured sparsity idea.

The paper is well written and easy to follow. The presentation of the algorithms for i) the non-overlapping and ii) overlapping groups as well as iii) the proposed refinement are clear. The experimental evaluation is interesting and convincing (the further experiments in the supplementary material add value to the overall discussion).

The main downside of the paper is that the proposed idea essentially consists in replacing the standard \ell_p norm penalty/constraints with a group-\ell_p one. While this provides interesting technical questions from the algorithmic perspective, from the point of view of the novelty, the paper does not appear an extremely strong contribution,

---

> ### Author Response · Authors · 2018-11-25
> **Response to Reviewer1**
>
> We thank the reviewer for the positive comments on our work. In addition to technical contributions from the algorithmic perspective, we would like to emphasize that StraAttackt identifies (group-wise) sparse adversarial patterns that make attacks successful, but without incurring extra pixel-level perturbation power compared to other existing attacks such as CW. The resulting sparse adversarial pattern also offers a visual explanation through adversarial saliency map (ASM) and class activation map (CAM). Effectiveness and interpretability of StrAttack reveals the ‘right’ pixels that an attacker should perturb to boost the attack performance. To strengthen this contribution, in the revised version we present the potential impacts of StrAttack from a) performance of attacking robust adversarially trained model (Table 2),  b) attack transferability (Table 3),  and c) interpretability of complex images (Figure 4).

---

### Official Review · AnonReviewer3 · 2018-11-08
**Interesting technical contribution**

**Rating:** 7
**Confidence:** 2

**Review:**

This paper proposes a method for adversarial attacks on DNNs (StrAttack), designed to exploit the underlying structure of the images. Specifically, incorporating group-sparsity regularization into the generation of the adversarial samples and using an ADMM based implementation to generate the adversarial perturbations.

The paper is structured and written well, with clear articulation of technical details. The experiments and reported results are comprehensive, and clearly showcase the efficacy of the proposed solution.  I'm not enough of an expert on the subject matter to comment about the novelty of this proposed approach. However, it would help to elaborate more on the related work (section.7) with clear contrasting of current method esp. using structural information for adversarial samples - theoretical implications, underlying rationale and importantly calling out the benefit over the previous lp - norm based approaches?

Regarding group sparsity - it is unclear as to the assumed structural constraints, is the sliding mask expected to be only 2x2, 13x13 (for MNIST/CIFAR-10, ImageNET respectively) ? impact of larger/smaller or skewed sizes ? sensitivity to image types?

---

> ### Author Response · Authors · 2018-11-25
> **Response to Reviewer3**
>
> We thank the reviewer for the positive comments, and answer your specific questions as below.
>
> a) In the revised version we have added more related work; see Introduction. Our structure-driven attack is motivated by devising a more efficient attack that takes advantages of two attacks using extremely opposite principles - C\&W attack (or \ell_infty attacks such as I-FGSM) that modifies all pixels, and one-pixel attack (Su et al., 2017) that only modifies a few pixels. The C\&W attack can achieve small \ell_infty perturbations but has to perturb most pixels (large \ell_0 norm). Although the one-pixel attack can achieve extremely small \ell_0 norm but with much higher \ell_infty norm and low attack success rate.  Both the above attack methods lead to higher noise visibility due to perturbing too many pixels or perturbing a few pixels too much. Motivated by them, we wonder if there exists a more effective attack that can be as successful as existing attacks but only requires to modify a small subset of pixels, and what the resulting sparse adversarial pattern can tell. To answer these questions, we propose StrAttack which achieves strong group sparsity without losing attack effectiveness including both attack success rate and Lp distortion. Furthermore, we show that the resulting sparse adversarial patterns offer a great interpretability through adversarial saliency map (ASM) and class activation map (CAM).
>
> b) The proposed StrAttack problem formulation cannot be solved using standard optimization solvers, e.g., Adam, or proximal gradient algorithm, etc, due to the presence of non-smooth regularizers and hard constraints. To address this technique challenge, we proposed the ADMM solution that splits the original complex problem into neat subproblems, each of which yields an analytical solution.
>
> c) We investigated the group sparsity by exploring various mask sizes. Clearly, there is a trade-off between the group size and the representation of local regions. Large mask size tends to make StrAttack insensitive to the structure of local regions. In the experimental evaluation of this paper, the best mask size that we empirically found are 2x2 MNIST/CIFAR-10 and 13x13 for ImageNET respectively.
>
> Last but not the least, to strengthen the effectiveness and the interpretability of StrAttack, in the revised version we present the potential impacts of StrAttack from a) performance of attacking robust adversarially trained model (Table 2),  b) attack transferability (Table 3),  and c) interpretability of complex images (Figure 4).

---

> > ### Comment · AnonReviewer3 · 2018-12-10
> > **RE: Response**
> >
> > Thanks for incorporating feedback, the additional related work section is helpful and provided better context for this work.

---

### Author Response · Authors · 2018-11-25
**General response**

We thank all reviewers for their insightful and valuable comments. Our paper has been greatly improved based on these comments. The major modifications are summarized as below.

a) We have enriched our related work and provided a better motivation on StrAttack (See Introduction).

b) To strengthen our contribution on the effectiveness of StrAttack, we have added experiments to show the attack performance of StrAttack against robust adversarial trained model [Madry et al. 2018]; see our results in Table 2. Moreover, we have compared the transferability of StrAttack with other attacks on 6 different network models (Table 3).

c) We have added more examples to show the better interpretability of StrAttack, where the found sparse adversarial patterns have a better correspondence with class-specific discriminative regions localized by CAM; see Figure 4.

Thanks!
ICLR 2019 Conference Paper689 Authors

---

> ### Comment · AnonReviewer2 · 2018-11-25
> **Thanks for the edits**
>
> Thank you for revising your paper, the new version seems to be more clear to me in terms if the positioning of your work. I have bumped up the numerical score to 6 in my review.

---

### Meta-Review · Area_Chair1 · 2018-12-17
**Accept.**

**Confidence:** 4
**Recommendation:** Accept (Poster)

**Metareview:**

This paper contributes a novel approach to evaluating the robustness of DNN based on structured sparsity to exploit the underlying structure of the image and introduces a method to solve it. The proposed approach is well evaluated and the authors answered the main concerns of the reviewers.